

# Swarm algorithms for sustainable dynamic flexible job shop rescheduling under machine breakdown in smart manufacturing plants

Nehal Tarek[1,2], Samia Allaoua Chelloug[3], Soha Alhelaly[4], Nancy A. El-Hefnawy[5], Hatem Abdel-Kader[1,2] and Amira Abdelatey[1,2]

[1] Department of Information Systems, Faculty of Computers and Information, Menoufia University, Shibin El Kom, Al Minufiyah, Egypt
[2] Faculty of Computers and Artificial Intelligence, Menoufia National University, Tukh Tambisha, Al Minufiyah, Egypt
[3] Department of Information Technology, College of Computer and Information Sciences, Princess Nourah bint Abdulrahman University, Riyadh, Saudi Arabia
[4] College of Computing and Informatics, Saudi Electronic University, Riyadh, Saudi Arabia
[5] Information Systems Department, Faculty of Computers and Informatics, Tanta University, Tanta, Egypt

Corresponding author
Nehal Tarek,
nehaltarek@ci.menofia.edu.eg

## ABSTRACT

Poultry manufacturing plants employing Dynamic Flexible Job Shop Scheduling Problems (DFJSSP) face workflow disruptions due to unexpected machine failures. Efficient rescheduling algorithms are essential to reallocate operations and minimize disruptions. This study proposes two machine failure handling strategies utilizing Grey Wolf Optimization (GWO) and Particle Swarm Optimization (PSO) techniques. Initially, the proposed algorithms generate a healthy state schedule that is executed and monitored in the manufacturing plant. In the event of a machine failure, the digital model of the system triggers a rescheduling process. In the first scenario, the finished operations are excluded out of the schedule, and the remaining operations are optimally rescheduled, excluding faulty machines. The results showed that GWO emphasizes aggressive makespan reduction but at higher energy costs, while PSO provides a more balanced trade-off with slightly longer makespan but lower energy consumption. In the second scenario, an operation-shifting technique is applied; disrupted operations are rescheduled on alternative machines, while following operations retain their initial assignments but are delayed. Both scenarios incorporate operational constraints and evaluate energy consumption to ensure efficiency. Accordingly, an energy consumption analysis report is provided to the decision makers to select the best scenario. The algorithm is implemented and tested under varying failure conditions. Both scenarios, with the proposed optimization algorithms, demonstrate effective rescheduling, with energy consumption analysis confirming rational energy use. As a confirmative step, the proposed GWO and PSO algorithms have been applied to the standard Brandimarte benchmark test cases with different problem sizes. The results proved the validity of the algorithms. Then, the system's performance has been investigated under different disruption times and failure scenarios. The proposed rescheduling algorithm proves robust and superior in handling machine failures. It minimizes workflow disruptions, ensures operational feasibility, and optimizes energy consumption, making it a reliable solution for poultry manufacturing plants. Finally, to validate the efficiency and effectiveness of

the proposed algorithm, we compare the outcomes of the two scenarios by applying them to the same problem and analyzing the behavior of each rescheduling strategy. Two powerful optimization techniques, GWO and PSO, were employed to assess the robustness of the rescheduling plan. Both techniques produced very similar results, demonstrating the consistency and reliability of the proposed approach.

# INTRODUCTION

## State-of-the art

In the context of smart manufacturing and in alignment with the "Artificial Intelligence and Localization of Industry in Egypt" strategy, enterprises strive for intelligent and precise production. Job shop scheduling problem (JSSP) is a control technology that can optimize production processes. By leveraging intelligent algorithms, production resources can be utilized efficiently despite operational constraints, resulting in significant cost reductions and a naturally shortened manufacturing cycle.

A digital twin (DT) is widely recognized as a virtual counterpart of a physical asset that replicates its behavior in real-time or near real-time. This virtual representation can encompass diverse entities such as vehicles, buildings, individuals, cities, organizations, or entire systems. Depending on its purpose, a DT may highlight specific dimensions like financial metrics or provide a comprehensive perspective that includes geographical configuration, human capital, asset inventory, and the interactions between various components. In recent years, the implementation of DT technology has accelerated, largely due to the rapid evolution of foundational technologies including big data, the Internet of Things (IoT), artificial intelligence and machine learning (AI-ML), and cloud computing. These technologies are vital for the precise acquisition, transmission, storage, and interpretation of the massive datasets generated by IoT devices. Consequently, DT has seen expansive deployment across various domains, with particularly notable impact in smart manufacturing (*Rosen et al., 2015*; *Xu et al., 2019*; *Qi & Tao, 2018*).

In real-world production systems, various dynamic events such as machine breakdowns (*Xiong, Xing & Chen, 2013*), uncertain processing times (*Chang & Liu, 2017*), order insertions (*Luo et al., 2020*), and operation inspections (*Zhu et al., 2023*) have been extensively studied by researchers. However, one of the most common dynamic events—order-related machine breakdowns—has not yet been thoroughly investigated, despite its significant impact on production scheduling. In *Zhu et al. (2023)*, a modified memetic algorithm was proposed for manufacturing systems that account for rework and scrap states of products. This approach integrates a hybrid scheduling method incorporating three distinct rescheduling strategies.

Over the past decade, several studies have explored anomaly detection algorithms in JSSP. In *Hu et al. (2023)*, an improved genetic algorithm (GA) with dynamic neighborhood search was introduced to address JSSP, focusing on minimizing the makespan. The authors of *Zheng et al. (2024)* examined the flexible JSSP (FJSSP) under uncertain processing times and proposed a robust optimization model utilizing a two-individual-based master–apprentice evolutionary algorithm. In *Li et al. (2023)*, an anomaly detection and dynamic scheduling framework based on DT technology was developed for flexible job shops, recognizing that sudden anomalies can cause scheduling deviations. To address this issue, the authors proposed a DT-based anomaly detection and dynamic scheduling method. Leveraging a rolling window mechanism, which extends heuristic algorithms to the dynamic scheduling domain, they introduced a time-window-nested multi-layer coding GA to optimize scheduling outcomes.

The FJSSP, first defined by *Brandimarte (1993)*, involves processing each operation on any machine within a specified subset of the total set of machines. When machine flexibility is high—meaning operations can be performed on multiple machines—finding high-quality or even optimal solutions become easier. As reviewed in *Dauzère-Pérès et al. (2024)*, various versions of the JSSP exist. In the basic JSSP, each job consists of a sequence of operations that have to be executed in a predetermined order, with each operation assigned to a specific machine. In the FJSSP variant, each operation of a job can be executed on a set of available machines. Further variations of the FJSSP include the partial FJSSP and the total FJSSP.

In the Dynamic FJSSP (DFJSSP), Jobs do not enter the workshop simultaneously at the start of the scheduling period; instead, they arrive at varying times throughout the process. Additionally, authors in *Dauzère-Pérès et al. (2024)* provided comprehensive insights into DFJSSP, covering criteria, constraints, additional problem characteristics, and solution methodologies. The DFJSSP extends the traditional job shop problem by introducing greater scheduling flexibility, but this also increases computational complexity, making it an NP-hard problem (*Xie et al., 2019*; *Zhong et al., 2020*).

*Sun, Cheng & Liang (2010)* proposed a GA with a penalty function for solving the JSSP. In *Gen, Tsujimura & Kubota (1994)*, the objective was to develop a GA-based approach for JSSP, demonstrating that even a relatively simple GA can effectively handle job shop scheduling. The research study in *Sun et al. (2019)* addressed FJSSP with uncertain processing times, represented by fuzzy numbers, and introduced a hybrid cooperative evolutionary algorithm to minimize the maximum fuzzy completion time. *Zhu et al. (2019)* applied an FJSSP model with combined processing constraints to the assembly manufacturing industry. In their study, the concepts of 'combined processing constraint' and 'virtual operation' were introduced to simplify and transform an FJSSP with combined processing constraints into a standard FJSSP. In *Stastny et al. (2021)*, a novel graph-based algorithm was proposed for optimizing scheduling problems. However, none of these studies specifically addressed anomaly operations.

In the DFJSSP, various constraints must be considered. In many cases, jobs are allowed to wait for only a limited time, as seen in semiconductor manufacturing, where excessive waiting can lead to product contamination, scrapping, or rework (*Mönch et al., 2011*).

Additionally, a minimum waiting time between two operations may be required, for example, to account for transportation delays between machines or to model processing times of operations that either do not require resources or rely on non-bottleneck machines (*Tamssaouet et al., 2022*). Instead of explicitly representing unavailability periods in the disjunctive graph, these periods can be incorporated by adjusting operation start times to prevent overlaps between processing and unavailability periods. This approach, as demonstrated in *Tamssaouet et al. (2022)*, effectively handles scheduling problems with fixed unavailability periods. In that study, operations were considered resumable under two scenarios; they could resume processing after being interrupted by an unavailability period.

The multiple resource-constrained JSSP is widely recognized as a typical NP-hard problem. To address this challenge, researchers frequently employ metaheuristic algorithms, including GA algorithms (*Tan et al., 2021*), Grey Wolf Optimization (GWO) algorithms (*Zhu et al., 2022*), Particle Swarm Optimization (PSO) (*Fontes, Homayouni & Gonçalves, 2023*), and differential evolution algorithms (*Wang, Gao & Pedrycz, 2022*). *Liu et al. (2023)* conducted an extensive study on the application of evolutionary algorithms to the fuzzy FJSSP with uncertain processing times. Their research focused on optimizing key performance metrics, such as maximum completion time, total machine load, and maximum machine load. To further enhance efficiency, they proposed a non-dominated sorting teaching-learning-based optimization algorithm aimed at minimizing both energy consumption and makespan.

## Research gap and motivations

In conclusion, while previous research has largely concentrated on minimizing machine energy consumption and makespan through multiobjective evolutionary algorithms, it has often neglected the inverse proportional relationship between time and energy. This oversight leads to a less realistic portrayal of actual production environments, where optimizing one objective often comes at the expense of the other, making it impractical to achieve simultaneous optimal minimization. Moreover, anomalous operations have frequently been ignored, limiting the applicability of these approaches in dynamic manufacturing contexts. A significant gap also remains in the literature regarding the use of GWO and PSO within DT-based DFJSSP, highlighting an important area for future exploration.

In this article, we propose a novel DFJSSP model that considers operation sequencing and timing while simultaneously addressing machine failure and redistribution scenarios. The key innovative contributions are as follows:

- A mathematical model for DFJSSP has been formulated in a DT system.
- GWO and PSO algorithms have been applied to minimize the maximum completion time (makespan).
- The dynamic rescheduling problem under machine breakdowns has been investigated, focusing on minimizing order completion time (makespan).

- Two breakdown handling scenarios have been proposed to ensure production continuity.
- Energy consumption has been analyzed and compared across three states: healthy state, failure state handled by Scenario 1, and failure state handled by Scenario 2, and then decide which scenario is better in terms of both makespan and rational use of energy.
- In the context of a DT system, sustainability is primarily reflected through continuous system monitoring, which enables early detection of machine failures and facilitates dynamic rescheduling. This ensures the stability and resilience of the production process, thereby reducing downtime, improving resource utilization, and minimizing waste—key factors contributing to a sustainable manufacturing environment.

The rest of this article is organized as follows. 'Related Work' summarizes the relevant literature in recent years. A description and the mathematical models of the system studied are presented in 'Industrial Manufacturing Plant Under Healthy and Failure States'. In 'Sustainable Dynamic Job Shop Rescheduling Problem Using GWO and PSO Optimization Algorithms', the two proposed DFJJSP scenarios are presented in detail *via* the proposed GWO and PSO algorithms. The simulation results and analysis are provided in 'Simulations and Analysis'. Finally, summary, limitations of the work, and prospects of the study are given in 'Conclusions'.

## RELATED WORK

Recently, considerable research interest has focused on the FJSSP when it accounts for machine breakdowns and urgent rush orders. To address the multi-objective DFJSSP, the authors in *Shen & Yao (2015)* proposed a proactive–reactive scheduling approach grounded in a multi-objective evolutionary algorithm. They further developed a dynamic decision-making framework to identify the optimal scheduling strategy. The research study (*Wang & Ding, 2020*) introduced a multi-objective differential evolution algorithm for dynamic FJSSP, considering both machine breakdowns and rush orders. Similarly, in *Baykasoğlu, Fatma & Alper (2020)*, the authors examined the dynamic FJSSP under uncertain conditions, including the arrival of new orders and machine breakdowns. They developed a constructive algorithm based on the greedy randomized adaptive search procedure to effectively address these dynamic disruptions.

In practical production scheduling, unforeseen events such as information asymmetry and abnormal disruptions often cause execution deviations, adversely affecting scheduling efficiency and quality. To tackle this issue, *Fang et al. (2019)* proposed an innovative job shop scheduling approach utilizing DT technology to better align scheduling plans with actual execution outcomes, particularly in the presence of dynamic random disturbances caused by uncertainties. They implemented a multi-objective optimization technique based on the nondominated sorting genetic algorithm (NSGA-II) to enhance scheduling performance. Nonetheless, the effectiveness of this method remains dependent on continuous updates from real-time data collected in the physical workshop.

Similarly, *Zhang, Tao & Nee (2021)* presents a dynamic scheduling strategy powered by DT technology. This approach uses a five-dimensional DT model tailored for a CNC

milling machine to predict machine availability, detect disturbances, and evaluate system performance. Based on this analysis, it determines whether delays in production (makespan) exceed a predefined threshold, prompting a rescheduling process. The main goals of this method are to maintain efficiency by minimizing makespan and to preserve stability by reducing shifts in operation start times.

While these studies focus on FJSSP under event-driven disruptions, they overlook the critical aspects of energy conservation and emission reduction. To align with green and sustainable development goals, energy consumption should be considered in future research.

*Mahmoodjanloo et al. (2021)* addressed a distributed job shop rescheduling problem in which facilities utilize reconfigurable machines. Initially, the problem was mathematically formulated to minimize total weighted lateness in a static state. Subsequently, the dynamic version was extended based on a conceptual rescheduling framework de-signed to update the current schedule. *Fu et al. (2024)* focused on scheduling a distributed flexible job shop with random job processing times to minimize both makespan and total tardiness. First, a stochastic programming model was developed to formulate the problem. Then, considering the dual-objective nature and inherent randomness, an evolutionary algorithm incorporating an evaluation method was designed. In *Cheng et al. (2024)*, the authors emphasized that mold changeover is often over-looked in multi-objective optimization and FJSSP. To address this issue, they designed an objective function that accounts for mold changeover time, providing valuable insights for multi-objective scheduling problems involving complex constraints. Rescheduling methods primarily include left-right shift rescheduling, partial rescheduling, and complete rescheduling. *Fuladi & Kim (2024)* presented a method for solving both static and dynamic FJSSP using a hybrid algorithm that combines GA, simulated annealing (SA), and variable neighborhood search. The method was tested on benchmark datasets and included a rescheduling strategy to handle dynamic events like machine breakdowns and job arrivals.

Swarm algorithms methods primarily include PSO, GWO, and ant colony optimization. For example, the work proposed in *Sha & Hsu (2006)* applied PSO combined with tabu search to solve the JSSP. In research focused on improving evolutionary algorithm performance, efforts have been directed toward enhancing exploration and exploitation capabilities. *Gao et al. (2020)* proposed an improved Jaya algorithm for the FJSSP with machine breakdowns and recovery, effectively addressing both constrained and unconstrained optimization problems. In *He et al. (2021)*, a dynamic integrated scheduling problem was investigated, considering breakdowns, order insertions, and the battery consumption of robots, with the objective of minimizing order completion time (makespan). Regarding FJSSP with rush orders, scholars have developed various metaheuristic optimization algorithms to enhance scheduling efficiency. *Gao et al. (2016)* proposed an improved artificial bee colony algorithm for the FJSSP with rush orders, considering fuzzy processing times. On the other hand, *Zhang et al. (2022)* introduced a hybrid approach combining variable neighbor-hood search and gene expression programming, incorporating four effective neighborhood structures. In *Zhu et al. (2024)*, the authors proposed a modified memetic algorithm to address job cancellations in

distributed FJSSP and examined solutions for job cancellations at various stages of job processing.

The GWO algorithm is a population-based intelligence algorithm originally designed to solve continuous optimization problems. It is inspired by the social hierarchy and hunting behaviors of grey wolves. Studies have demonstrated that GWO yields competitive results compared to well-known metaheuristic algorithms. A more advanced implementation of the GWO algorithm has been developed for the FJSSP, incorporating job redistribution and machine reassignment following machine breakdowns. GWO has been widely applied across various fields, including optimization, classification, economic and power dispatch, and capacitated vehicle routing (*Jiang & Zhang, 2018*). *Jiang et al. (2018)* applied an improved GWO algorithm with two search modes to solve the JSSP. *Lu et al. (2017)* tackled a complex and practical problem of dynamic scheduling in a real-world welding industry. Their study accounted for key dynamic events such as machine breakdowns, job quality issues, and job release delays. To address the resulting multi-objective dynamic scheduling problem, they proposed a hybrid multi-objective GWO aimed at minimizing makespan, machine load, and schedule instability. Their approach demonstrated strong applicability to dynamic industrial environments, highlighting the potential of GWO-based algorithms in practical scheduling contexts. A tri-objective hybrid flowshop scheduling problem (HFSP) with controllable processing times was investigated by authors of *Lu et al. (2019)* to minimize the makespan, noise pollution and energy consumption.

In *Chen, Chou & Chou (2020)*, a multi-objective evolutionary approach was proposed to handle integrated airline scheduling and rescheduling problems under disruption conditions. To manage disturbances in FJSSP involving automated guided vehicle transportation, a mixed-integer linear programming model was developed (*Zhang et al., 2023*). Based on the characteristics of this model, an improved NSGA-II algorithm was designed to minimize makespan, energy consumption, and machine workload deviation. *Kong et al. (2022)* proposed a discrete improved GWO algorithm specifically designed for FJSSP, incorporating hybrid initialization strategies and adaptive convergence factors. In *Li et al. (2022)*, novel encoding and decoding schemes were introduced to represent subproblems and transform them into feasible schedules, thereby enhancing the effectiveness of the GWO algorithm. Additionally, *Zhou et al. (2024)* developed an adaptive GWO algorithm that dynamically selects between global and local search strategies based on the degree of individual agglomeration, improving optimization precision and convergence speed.

The PSO is a population-based optimization algorithm inspired by the social behavior of birds or fish. It uses a group of candidate solutions, called particles, which move through the search space to find the optimal solution. Each particle adjusts its position based on its own experience and the experience of the best-performing particle in the swarm. Over time, particles converge toward the best solution found. PSO is widely used for solving optimization problems due to its simplicity, efficiency, and ability to handle complex, nonlinear search spaces.

*Kong & Wang (2024)* proposed discrete particle swarm algorithm to consider handling and setup time. First, a multi-objective optimization model was developed, focusing on minimizing the maximum completion time, the total number of machine adjustments, the total number of workpiece handlings, and the overall machine load. Then, an enhanced discrete particle swarm optimization algorithm was proposed, incorporating Pareto-based selection and an adaptive nonlinear strategy for adjusting inertia weight. *Xu et al. (2025)* explored the use of Quantum Particle Swarm Optimization (QPSO) enhanced with chaotic encoding schemes to solve the FJSSP. Fourteen chaotic maps were evaluated within the QPSO framework, showing improved solution quality and faster convergence. The findings highlight the potential of combining quantum and chaos theories for more effective scheduling optimization.

*Ngwu, Liu & Wu (2025)* reviewed AI-based methods for DJSSP, focusing on how reinforcement learning (RL) adapts to unpredictable events like job arrivals and machine failures. It highlighted RL's strengths in managing complex, real-time scheduling environments and identifies current challenges such as scalability and data limitations. The study in *Wu, Zheng & Yin (2025)* proposed a dual-objective deep reinforcement learning approach using Double Deep Q-Networks (DDQN) with attention mechanisms to handle DFJSSP under machine breakdowns. Additionally, *Chang, Liu & You (2024)* addressed the DFJSSP by proposing an improved learning-to-dispatch model using graph neural networks and RL. It formulated scheduling as a disjunctive graph to handle changing machine availability. *Albayrak & Onuet (2024)* addressed sustainable manufacturing by tackling the multi-objective FJSSP with a focus on energy efficiency, machine workload, and makespan. It incorporated dynamic events like new job arrivals and rework processes to reduce scrap. An enhanced NSGA-II algorithm was used to solve the problem.

Table 1 summarizes a comparison between all the above-mentioned research efforts. The first column indicates the reference number, the second column specifies the type of problem studied, the third column defines the objectives to be minimized. Finally, in the last column, the optimization algorithm, that is used in relevant works, is presented.

# INDUSTRIAL MANUFACTURING PLANT UNDER HEALTHY AND FAILURE STATES

## System framework architecture

This manuscript is an extension of the work presented in reference *Tarek et al. (2025)*. The proposed system framework is designed to be compatible with small- to medium-sized manufacturing plants. The case study focuses on a poultry manufacturing plant comprising three production lines. Each line includes seven machines that manage the production process—from receiving raw material containers to delivering the final product in sewed packages. Further details are available in *Tarek et al. (2025)*. The architecture of the proposed framework, including the rescheduling optimization component, is illustrated in Fig. 1. It consists of three main modules: the input data module, the digital twin module, and the optimization module. The input data module initializes the system using machine data (historical and real time sensor data), objectives, constraints, available

**Table 1  An overview of the literature review.**

| Reference number | Job shop type | Objective function | Optimization algorithm |
|---|---|---|---|
| *Shen & Yao (2015)* | DFJSSP | Makespan, machine workload balance, total tardiness | Multi-objective evolutionary algorithms (MOEAs) |
| *Wang & Ding (2020)* | DFJSSP | Makespan, total tardiness, machine load balance. | Improved GA |
| *Baykasoğlu, Fatma & Alper (2020)* | DFJSSP | Mean tardiness, schedule instability, makespan, mean flow time. | Greedy randomized adaptive search procedure |
| *Fang et al. (2019)* | DT based JSSP | Makespan, scheduling robustness, resource utilization | NSGA-II |
| *Zhang, Tao & Nee (2021)* | DT based DJSSP | Makespan, changes of starting time of each operation in the rescheduling | Neural network |
| *Fu et al. (2024)* | FJSSP | Makespan, total tardiness | MOEA |
| *Cheng et al. (2024)* | FJSSP | Makespan, setup time | Improved GA |
| *Fuladi & Kim (2024)* | DFJSSP | Makespan | GA integrated with SA |
| *Sha & Hsu (2006)* | JSSP | Makespan | PSO, Tabu search |
| *Gao et al. (2020)* | FJSSRP | Makespan, instability | Improved jaya algorithm |
| *Gao et al. (2016)* | FJSSP | Makespan | Artificial bee colony |
| *Zhang et al. (2022)* | FJSSP | Makespan | GA |
| *Zhu et al. (2024)* | FJSSP | Makespan, energy consumption | Reformative memetic algorithm |
| *Jiang & Zhang (2018)* | FJSSP | Makespan | GWO |
| *Jiang et al. (2018)* | JSSP | No-load energy consumption cost, the total tardiness cost | GWO |
| *Lu et al. (2017)* | DJSSP | Makespan, penalty of machine load, instability | Hybrid multi-objective GWO |
| *Lu et al. (2019)* | Flow shop scheduling problem | Makespan, noise pollution, energy consumption | Cellular GWO |
| *Zhang et al. (2023)* | FJSSP | Makespan, energy consumption, sum of workload deviation | Improved NSGA-II |
| *Kong et al. (2022)* | FJSSP | Makespan, workload | GWO |
| *Zhou et al. (2024)* | FJSSP | Makespan | GWO |
| *Kong & Wang (2024)* | FJSSP | Makespan, machine adjustments, workpiece handlings, load of the machine | PSO |
| *Xu et al. (2025)* | FJSSP | Makespan | PSO |
| *Wu, Zheng & Yin (2025)* | DFJSSP | Total tardiness, machine offset | Double deep Q-network |
| *Chang, Liu & You (2024)* | DFJSSP | Makespan | Learning-to-dispatch reinforcement learning |
| *Albayrak & Onuet (2024)* | DFJSSP | Makespan, energy consumption, machine workload. | NSGA-II |

resources, and a comprehensive list of jobs and operations needed to fulfill product requirements. The digital twin module integrates machine availability, machine state (healthy or in breakdown), and the simulation module. Then, send all data to the optimization module. The initial schedule is optimized using the GWO and then send the resulting schedule back to DT.

The next phase is starting the execution of the production process; if a machine fails, DT module sends a failure report and then the system triggers rescheduling based on one of two predefined scenarios (Scenario 1 or Scenario 2). Table 2 provides a summary of the

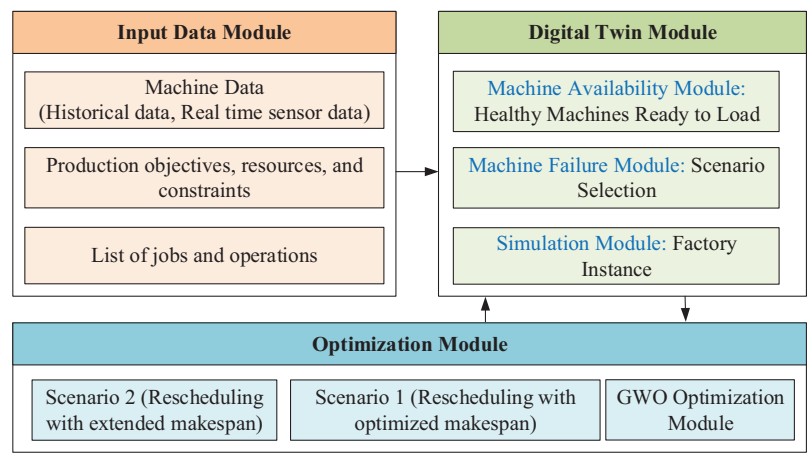

**Figure 1** DT-based framework architecture.

**Table 2** The production operations and the corresponding processing machines and processing time.

| Operation number | First production line time | | Second production line machine time | | Third production line machine time | |
|---|---|---|---|---|---|---|
| | Machine | Time | Machine | Time | Machine | Time |
| $O_{j1}$ Milling | $M_1$ | 18 | $M_8$ | 6 | $M_{15}$ | 4 |
| $O_{j2}$ Mixing | $M_2$ | 24 | $M_9$ | 18 | $M_{16}$ | 12 |
| $O_{j3}$ Adding liquid | $M_3$ | 8 | $M_{10}$ | 6 | $M_{17}$ | 4 |
| $O_{j4}$ Pelletizer | $M_4$ | 12 | $M_{11}$ | 4 | $M_{18}$ | 3 |
| $O_{j5}$ Cooler | $M_5$ | 6 | $M_{12}$ | 2 | $M_{19}$ | 2 |
| $O_{j6}$ Scaling | $M_6$ | 2 | $M_{13}$ | 1 | $M_{20}$ | 1 |
| $O_{j7}$ Sewing | $M_7$ | 2 | $M_{14}$ | 2 | $M_{21}$ | 1 |

processing times for all operations across the corresponding machine set. Processing time is measured in time units, where each unit corresponds to 5 min. This case study considers ten jobs, each comprising seven operations, with each operation executable on the corresponding machine within any of the three production lines.

## Simulation environment with failure state

The simulation environment in case of failure in the three production lines poultry factory, as defined in *Tarek et al. (2025)*, is illustrated in Fig. 2. This figure demonstrates the core concept of rescheduling job shop operations when certain machines are turned off. When a machine fails, its operations are redistributed among alternative healthy machines to maintain workflow continuity.

- Line 1: Machine M1 is in a failure state, and its operations are reassigned to M8 and M15.
- Line 2: Machine M10 experiences a failure, with its operations redistributed between M3 and M17.
- Line 3: Machine M19 is in a failure state, and its operations are reassigned to M5 and M12.

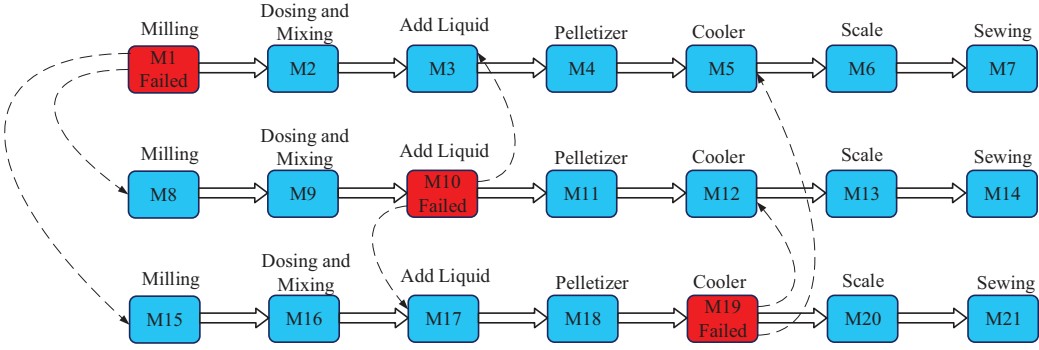

**Figure 2** Smart poultry feed planet (SPFP) graph with three failed machines.

Operations assigned to failed machines (marked in red in Fig. 2) are referred to as disrupted operations. These operations are followed by subsequent operations, which are called following operations and may require different rescheduling strategies. The specific redistribution scenarios for disrupted and following operations are detailed in the next section.

## DFJSSP problem formulation

To effectively address the DFJSSP, the following key assumptions, objectives, and constraints are considered:

### Solving a DFJSSP involves three primary decisions

- Machine assignment—determining which machine will process each operation.
- Operation sequencing—defining the order of operations within a job.
- Operation timing—scheduling the start and end times of operations.

### Key assumptions

- The sequence of operations within the same job is fixed, while there are no sequence constraints between different jobs.
- Each machine is designated for a specific type of machining process and can handle only one task at a time.
- Once the operation starts, it must run to completion unless interrupted by special circumstances such as machine breakdowns or order cancellations.

Additional assumptions are detailed in *Tarek et al. (2025)*. For clarity, the notations used in the mathematical model are listed in Table 3.

The fitness function for the scheduling problem aims to minimize the makespan—the maximum completion time $Max_{1 \leq j \leq n}(CT_j)$ for all jobs, as defined in Eq. (1). This optimization is subject to the following constraints: Constraint (2) (Eq. (2)) ensures that each operation is assigned to only one machine from the available set. Constraint (3) (Eq. (3)) guarantees that at any given time $t$, a machine can process only one operation.

**Table 3 Notations used in the mathematical model.**

| Notation | Description |
|---|---|
| $n$ | Number of jobs |
| $m$ | Number of machines |
| $l$ | Number of operations in a single job |
| $J_j$ | the $j$th job where $1 \le j \le n$ |
| $O_{ji}$ | The $i$th operation of the $J_j$ where $1 \le i \le l$ |
| $O_{jid}$ | Disrupted operation |
| $O_{jif}$ | Finished operation |
| $O_{jie}$ | Executing operation |
| $O_{j(i+1)fl}$ | Following operation for disrupted operation $O_{jid}$ |
| $M_k$ | The $k$th machine where $1 \le k \le m$ |
| $FM$ | List of failure machines |
| $AM_i$ | List of alternative machines for operation i, $1 \le i \le l$ |
| $DO$ | List of disrupted operations |
| $FO$ | List of finished operations |
| $EO$ | List of executing operations |
| $RO$ | List of remaining operations |
| $T_F$ | Failure time |
| $P_{ji}$ | The processing time of $O_{ji}$ |
| $P_{jik}$ | The processing time of $O_{ji}$ on machine $k$ |
| $EC_k$ | The energy consumption of machine $k$ where $1 \le k \le m$ |
| $ST_{ji}$ | The starting time of operation $O_{ji}$ |
| $ETO_{ji}$ | The ending time of operation $O_{ji}$ |
| $STO_{j(i)kfl}$ | The start time of the following operation $O_{j(i)}$ on machine $k$ |
| $NSTO_{j(i)kd}$ | The new start time of the disrupted operation $O_{j(i)d}$ on machine $k$ |
| $NETO_{j(i)kd}$ | The new end time of the disrupted operation $O_{j(i)d}$ on machine $k$ |
| $BMO_{j(i)d}$ | The selected machine to process the disrupted operation $O_{j(i)}$ |
| $CT_j$ | The completion time of the $J_j$ |
| $CT_k$ | The completion time of the $M_k$ |
| $P_{avk}$ | The average power for a machine $k$ |
| $RT_k$ | The running time for a machine $k$ |
| $EC_k$ | The energy consumption for a machine $k$ |
| $TEC$ | The total energy consumption in the production cycle |
| $x_{jik}$ | $= \begin{cases} 1 & \text{if operation } O_{ji} \text{ is assigned to machine } k \\ 0 & \text{otherwise} \end{cases}$ |
| $x_{jikt}$ | $= \begin{cases} 1 & \text{if operation } O_{ji} \text{ is assigned to} \\ & \text{machine } k \text{ on time } t \\ 0 & \text{otherwise} \end{cases}$ |
| $xo_{jidk}$ | $= \begin{cases} 1 & \text{if disrupted operation } O_{jid} \text{ is assigned to} \\ & \text{machine } k \\ 0 & \text{otherwise} \end{cases}$ |

Constraint (4) (Eq. (4)) enforces the precedence rule, ensuring that operations within the same job follow the correct sequence.

$$Min\left(Max\left(CT_j\right)\right) \qquad for\ 1 \le j \le n. \tag{1}$$

Subject to

$$\sum_{k=1}^{m} x_{jik} = 1 \ \forall\ j \in \{1, 2, \ldots, n\},\ \forall\ i\ \in \{1, 2, \ldots, l\}. \tag{2}$$

$$\sum_{j=1}^{n} \sum_{i=1}^{l} x_{jikt} \le 1 \ \forall\ k \in \{1, 2, \ldots, m\}. \tag{3}$$

$$ST_{ji} + P_{ji} \le ST_{j(i+1)} \ \forall\ j \in \{1, 2, \ldots, n\},\ \forall\ i\ \in \{1, 2, \ldots, l\}. \tag{4}$$

## SUSTAINABLE DYNAMIC JOB SHOP RESCHEDULING PROBLEM USING GWO AND PSO OPTIMIZATION ALGORITHMS

In this work, our primary objective is to minimize the makespan, as it is a critical performance indicator in many real-world scheduling scenarios. Incorporating additional objectives, such as stability or robustness, could provide a more comprehensive optimization approach. However, incorporating multiple objectives would inevitably involve trade-offs and may reduce the degree of optimality achievable for makespan minimization due to the required weighting and balancing among objectives. To partially address this concern, we evaluated, at the end of this section, the energy consumption associated with both the GWO and PSO under various scheduling scenarios. This evaluation is intended to demonstrate that the solutions obtained not only achieve minimized makespan but also remain within acceptable operational limits for energy usage.

### Scheduling and rescheduling strategy

Emergencies such as machine breakdowns and rush orders significantly impact the production activities of manufacturing enterprises. The dynamic scheduling strategy employs an event-based rescheduling approach, focusing on assessing the state of all operations at the time of failure. It then modifies these states to ensure production continuity by determining job positions, identifying unprocessed tasks, and calculating the earliest available time for machines. This article specifically examines one type of dynamic event: machine breakdowns, which can occur at any point during the production process. When a machine breaks down, the operation being processed on the faulty machine is scrapped.

The extended makespan time varies depending on the severity of the fault, as production lines differ in terms of machine capacity and power. The same operation can be processed on alternative machines, but with varying processing times, as illustrated in Table 2. Normal machines that remain operational are rescheduled either immediately or after the completion of their current processes. To address disruptions in DFJSSP with

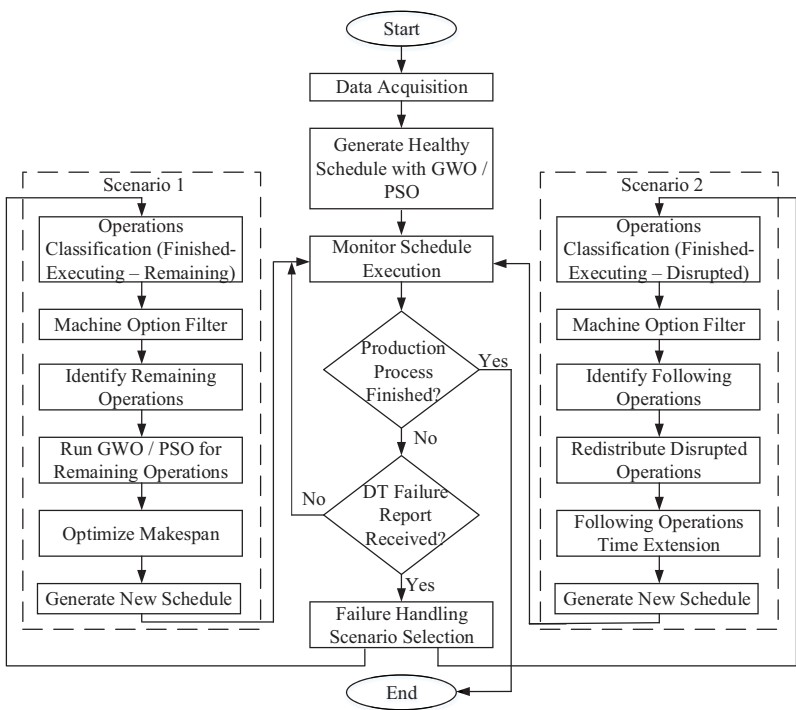

**Figure 3 Flow chart for the production process with dynamic scheduling to handle machine failure.**

automated guided production lines, a GWO and PSO model is developed to minimize the makespan.

The dynamic scheduling of the production flow, based on operation inspection, is illustrated in Fig. 3. The flowchart outlines a process starting with data acquisition, followed by GWO and PSO optimization. Afterward, a model or rule execution step is performed. The next decision point checks whether the production process is finalized. If yes, the process ends. If not, another decision checks if a DT failure report has been received. If no report is received, the process continues with the initial schedule. But if a failure report is received, a scenario selection step is performed which leads to two possible scenarios:

### Scenario 1

- Operations until the time of failure are classified as: Finished, Executing, and Disrupted operations. Where finished operations are the operations whose end time is before the failure time. Executing operations are the operations whose start time is before failure time and its end time is after failure time. Considering that executing operations are on healthy machines only. Disrupted operations are the operations whose start time is before failure time and its end time is after failure time, but only if it was assigned to a failed machine.
- Machine options are filtered to exclude the failed machines from the set of alternatives.

- Remaining operations are identified in this step, and it is defined by the operations which are not finished operations neither executing operations.
- The GWO optimization is applied to the remaining operations.
- The model is optimized.
- Generate new schedule.

### Scenario 2

- Operations until the time of failure are classified (the same as Scenario 1).
- Machine options are filtered (the same as Scenario 1).
- Following operations are identified and removed from the initial schedule. If a disrupted operation is determined, assuming it is the fifth operation in a certain job, then the sixth and seventh operations within the same job are identified as following operations. And this is repeated for all disrupted operations.

- Redistribute disrupted operations on the other two alternative machines.
- The following operations are reassigned to the same machines originally assigned to them, but their start times are delayed aligning with either the machine's next available time (after completing all initially scheduled operations) or the end time of the preceding operation.
- Generate new schedule.

Finally, after completing either of the scenarios, the new schedule is fed back to the DT module to resume the production process and continue monitoring the plant.

### Initial scheduling

In this step, the scheduling of the operations is distributed according to the proposed GWO optimization, as shown in Fig. 3. Here, the machine availability will be recorded to be used in the after-failure scenarios. The steps involved in this algorithm can be summarized as follows:

### Problem setup

This step involves acquiring data related to jobs and operations, available machines for each operation, machine processing times, and GWO optimization parameters (*i.e.*, number of wolves and number of iterations). It also stores fitness functions and system constraints.

### Initialization

1. Population initialization:

The population of wolves (solutions) is initialized. Each wolf represents a schedule, which is a list of operations with assigned machines and processing times in the following form (job number, operation number, assigned machine, processing time, start time, end time).

2. Alpha, Beta, Delta wolves:

The alpha, beta, and delta wolves (representing the top three solutions) are initialized to none.

### Scheduling optimizer

**A) Grey Wolf Optimization**

The algorithm runs for 100 iterations, in each iteration:

1. Fitness evaluation:
   - The fitness (makespan) of all wolves is evaluated as in equation $Max_{1 \leq j \leq n}(CT_j)$ in Eq. (1).
   - The fitness values and modified schedules are stored.

2. Sort volves by fitness:
   - Wolves are sorted by their fitness values (lower makespan is better).
   - The alpha, beta, and delta wolves are updated to the top three wolves in the sorted list.

3. Update positions of wolves:
   - For each wolf, the distances between the wolf and the top three wolves (alpha, beta, and delta) are calculated according to the next Eqs. (5) through (7).

$$D_\alpha = \left| C_1 \, X_\alpha - X_{(t)} \right|. \tag{5}$$

$$D_\beta = \left| C_2 \, X_\beta - X_{(t)} \right|. \tag{6}$$

$$D_\delta = \left| C_3 \, X_\delta - X_{(t)} \right|. \tag{7}$$

Here, $D_\alpha$, $D_\beta$, and $D_\delta$ represent the distances between the current wolf and the top three solutions (alpha, beta, and delta). The coefficients $C_1$, $C_2$, and $C_3$ are randomly generated using the formula $C = 2r_1$, where $r_1$ is a random vector that can take values of either 0 or 1. $X_\alpha$, $X_\beta$, and $X_\delta$ denote the machine assignments for the alpha, beta, and delta wolves, respectively, while $X_{(t)}$ represents the machine assignment for the wolf in the current iteration.
   - The GWO equations are used to calculate the new position of each wolf as in Eq. (8).
   - The new position is clamped to the nearest available machine option.

$$X_{(t+1)} = \frac{1}{3} \left[ (X_\alpha - A_1 \, D_\alpha) + (X_\beta - A_2 \, D_\beta) + (X_\delta - A_3 \, D_\delta) \right]. \tag{8}$$

Here, $X_{(t+1)}$ represents the updated position of the wolf, calculated as the average of its movements toward the alpha, beta, and delta wolves. The coefficients $A_1$, $A_2$, and $A_3$ are randomly determined using the formula $A = 2r_2 a - a$, where aa is a parameter that decreases linearly from 2 to 0 over the course of iterations (facilitating the transition from exploration to exploitation), and $r_2$ is a random value that can be either 0 or 1. This equation determines the new machine assignment $X_{(t+1)}$ for the wolf by averaging the three directional movements based on its distances to the alpha, beta, and delta wolves.

4. Update best makespan:
- The list of solutions is sorted and the first top three are assigned to alpha, beta, and delta.
- The makespan of the alpha wolf (best solution) is updated.
- The makespan of this iteration is compared with the previous one to store the minimum of them.

**B) Particle Swarm Optimization**

The PSO optimization algorithm can be described in the following steps:

1. Particle initialization
- Each particle represents a potential schedule (position) using a list of job-operation-machine-time.
- Machines are randomly selected from the options for each operation.
- A velocity is assigned to each operation, initialized randomly in a range.

2. PSO initialization
- A swarm of particles is created.
- Global best position and makespan are initialized with infinite makespan.
- PSO parameters (w, c1, c2) are defined to control exploration and exploitation.

3. Iterative optimization loop

The algorithm runs for 100 iterations, in each iteration:

a. Evaluate each particle
- For each particle, calculate the makespan of its current position as in equation $Max_{1 \leq j \leq n}(CT_j)$ in Eq. (1).
- Update the particle's personal best if the current makespan is better.
- Update the global best if this particle's makespan is better than the current global best.

b. Update velocity
- For each particle:
  - Determine the current, personal best, and global best machine indices.
  - Compute new velocity based on inertia, personal influence, and social influence as in Eq. (9).
  - Clamp velocity within a fixed range to ensure stable updates.

$$v_i = \omega v_i + c_1 r_1 (p_i - x_i) + c_2 r_2 (g_i - x_i) \tag{9}$$

where

$\omega$ inertia weight

$c_1$ cognitive coefficient

$c_2$ social coefficient

$v_i$ the current velocity for operation $i$

$x_i$ current machine index,

$p_i$ personal best machine index

$g_i$ global best machine index

$r_1$, $r_2$ generated random numbers $\in [0,1]$

c. Update position

- Based on the new velocity, update the particle's machine choices for each operation according to Eq. (10).

$$x_{i(new \text{ machine index})} = x_{i(current \text{ machine index})} + v_i. \tag{10}$$

### Generating Gantt chart

In this step, the system generates Gantt chart for the best optimized schedule after the GWO and the PSO algorithms terminate and then send it to the DT module for monitoring.

### Monitor schedule execution

At this stage, the DT module continuously tracks the progress of the execution of the optimized schedule and sensing for any machine anomaly behavior inside the manufacturing plant. It also Identifies the current state of operations (*e.g.*, finished, executing, or yet to start).

## Scenario 1: dynamic flexible rescheduling with optimized makespan

This scenario focuses on addressing machine failure during the execution of the optimized schedule generated by the optimization algorithm. This dynamic event is triggered when a machine failure situation is detected at a certain time, as reported by the DT monitoring module, which provides also a list of failed machines. The steps of the machine failure handling involved in this scenario are illustrated in this section. The algorithm in this scenario works as follows:

1. Classifying operations:

At the time of failure, Adding each operation to one of the following categories:

- Finished operations (operations completed before the failure time).
- Executing operations (ongoing operations at the failure time).
- Disrupted operations (ongoing operations on failed machines at failure time).

2. Filtering machine options:

- This step removes failed machines from the alternative machine set for each operation. And adjust the corresponding processing time.

3. Filter remaining operations:

- This step removes both finished operations and executing operations from the total operations.

4. Redistribute remaining operations using both GWO and PSO:
   - Sending the remaining operations as input dataset for the GWO and the PSO algorithm considering the start time is the machine failure time. And check the availability of machines to see if one of the executive operations is still running.

5. New schedule and makespan:
   - The new schedule and its makespan are calculated and stored.

6. Visualize the updated schedule:
   - Generate a Gantt chart to visualize the updated schedule, highlighting the changes caused by the machine failure and the redistribution process.

This information is fed back to DT module to update and start monitoring the execution of the updated schedule. The pseudocode for GWO and PSO algorithms are presented in Figs. 4 and 5, respectively.

Figure 6 provides an example illustrating how disrupted operations can be rescheduled using Scenario 1 approach. In this scenario, machine M10 is assumed to fail, requiring the redistribution of operations op(1-3), op(3-3), op(4-3), op(6-3), and op(8-3) to alternative machines. According to Fig. 2, M3 and M17 serve as alternatives for M10. Before the machine failure, the optimizers generated the scheduling using all machines (M1–M21). Once M10 experiences a breakdown, the optimizer identifies and manages the disrupted operations after filtering out the faulty machine. The remaining operations are then optimally rescheduled among the functional machines.

For instance, as illustrated in Fig. 6, machine M10 fails at time 20. The disrupted operation op(1-3) is reassigned to M17 by the optimizer, while op(3-3) is executed by M3 after the completion of op(2-3). Similarly, op(4-3) is moved to M17, op(6-3) to M3, and op(8-3) also to M3. Following the failure, the optimization algorithm—whether using GWO or PSO—redistributes operations from the functioning machines to suitable alternatives, ensuring minimal disruption and maintaining scheduling efficiency.

## Scenario 2: dynamic flexible rescheduling with extended makespan

Scenario 2 focuses on addressing machine failures during the execution of the optimized schedule generated by the optimization module. During the machine failure handling, the dynamic event is triggered when a machine failure situation is detected at a certain time, as reported by the DT monitoring module, which provides also a list of failed machines. The algorithm in this scenario works as follows:

1. Classifying operations:

At the failure time, adding each operation to one of the following categories:

- Finished operations (operations completed before the failure time).
- Executing operations (ongoing operations at failure time).
- Disrupted operations (ongoing operations on failed machines at failure time).

```
Initialize variables
GWO
   Initialize wolves
   for each iteration:
      for each Wolf:
         calculate makespan:
            initialize: machine_available_time, job_available_time
            for each operation in each job:
               check precedence constraint
               calculate: start time, end time
               update machine and job availability
            makespan = max of job_available_time
            end for
      end for
      sort wolves by fitness (lower is better)
      update Alpha, Beta, Delta wolves
      for each wolf:
         for each job:
            for each operation:
               calculate coefficients A and C
               retrieve machine assignments
               calculate new position
               clamp to available machine options
            end for
         end for
      end for
      calculate current best makespan
      update best makespan if current is better
   return final best schedule and makespan
if failure:
   get finished_operations, executing_operations, disrupted_operations
   get new_machine_options, new_processing_times
   get remaining_operations
   go to GWO
```

**Figure 4 The Pseudo code for GWO of healthy case and Scenario 1.**

2. Filtering machine options:
   - This step removes failed machines from the alternative machine set for each operation.

3. Identifying following operations:
   - This step identifies all operations that follow disrupted operations in the same job. For example, if operation number 4 is marked as disrupted operation, so operations number 5, 6 and 7 are all marked as following operations within the same job.

```
Initialize variables
PSO
   Initialize particles
   for each iteration:
      for each particle:
         calculate makespan:
         if makespan < particle.best_makespan:
            update particle best makespan
            update particle best schedule
         if makespan < global_best_makespan:
            update global best makespan
            update global best schedule
      end for
      for each particle:
         calculate cognitive coefficient
         calculate social coefficient
         update velocity
         update position
   return final best schedule and makespan
if failure:
   get finished_operations, executing_operations, disrupted_operations
   get new_machine_options, new_processing_times
   get remaining_operations
   go to PSO
```

**Figure 5** **The pseudocode for PSO of healthy case and Scenario 1.**

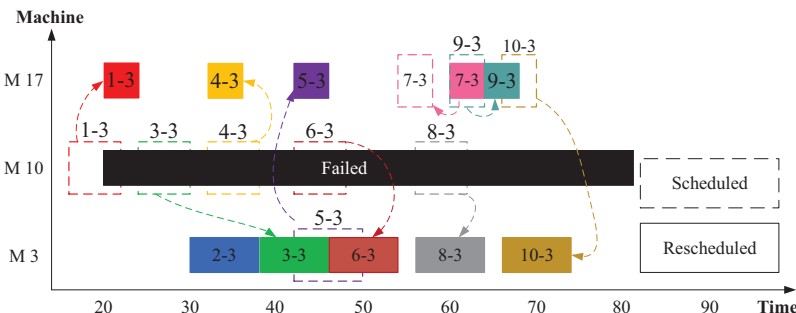

**Figure 6** **Rescheduled distribution of jobs and operations in Scenario 1.**

4. Redistribute disrupted operations:
   • At this point, the algorithm reassigns disrupted operations to alternative machines while respecting precedence constraints and machine availability time according to Eq. (11).

$$\sum_{k=1}^{m} xo_{jidk} = 1 \quad \forall\, o_{jid} \in \{DO\},\; \forall\, k \in \{AM_i\} \text{ and } k \notin \{FM\}. \tag{11}$$

5. Reassign following operations:
   - After redistributing all disrupted machines to the alternative machines, the following operations are reassigned to their initial assigned machines but after extending the start time of the operation to the maximum of either the machine's availability time or the completion time of the precedence operation within the same job. This expressed in Eq. (12).

$$STO_{j(i)kfl} = Max\big(ETO_{j(i-1)}, CT_k\big) \tag{12}$$

   where $STO_{j(i)kfl}$ is the start time of operation $O_{j(i)}$, which is originally assigned to machine $k$ and follows a disrupted operation. $ETO_{j(i-1)}$ is the end time of the preceding operation within the same job, and $CT_k$ is the availability time of machine $k$.

6. New schedule and makespan:
   - The new schedule and its makespan are calculated and stored.

7. Visualize the updated schedule:
   - Generate a Gantt chart to visualize the updated schedule, highlighting the changes caused by the machine failure and the redistribution process.
   - This information is fed back to DT module to update and start monitoring the execution of the updated schedule

Scenario 2 does not involve the use of optimization techniques. Instead, it relies on a shifting mechanism to reassign the disrupted operation to one of the available alternative machines. Subsequently, all the following operations are shifted accordingly to maintain the precedence constraints. As a result, Scenario 2 does not require the application of GWO or PSO. The pseudocode, illustrating the implementation of Scenario 2 and the shift methodology, is presented in Fig. 7.

First, as a case study, we explain here the operation of shifting algorithm after machine M10 fails. First, the disrupted operations, that were executed by the failed machine (op(1-3), op(2-3), and op(9-3)), are rescheduled on the related alternative machines M3 and M17. To decide which machine to select, the algorithm tests the two alternative machines. The machine with the earliest end time is chosen. Figure 8 shows the rescheduling process for these three disrupted operations. The new start time of the disrupted operation op(1-3) is selected as the maximum value between the end time of the previous operation, failure time, and machine availability time as in Eq. (13).

$$NSTO_{j(i)kd} = Max\big(ETO_{j(i-1)}, T_F, CT_k\big) \tag{13}$$

where $NSTO_{j(i)kd}$ is the new start time of the disrupted operation $O_{j(i)}$ on machine $k$, selected from the alternative machine set. $T_F$ is the machine failure time.

```
if failure:
    get finished_operations, executing_operations, disrupted_operations
    get new_machine_options, new_processing_times
    get following_operations
    get_machine_availability
    for each disrupted_operation:
        for each alternate_machine:
            get previous_operation
            if previous_operation:
                new_start_time = max(previous_operation, time_unit,machine_availability)
            else:
                new_start_time = max(time_unit, machine_availability)
            new_end_time = new_start_time + processing_time
        end for
        select minimum end_time
        update operation info
    end for
    for each following_operation:
        get previous_operation
        if previous_operation:
            start_time = max(prev_op, machine_availability)
            end_time = start_time + processing_time
            machine_availability = end_time
    calculate makespan
    return final schedule and makespan
end
```

**Figure 7 The pseudocode for Scenario 2.**   

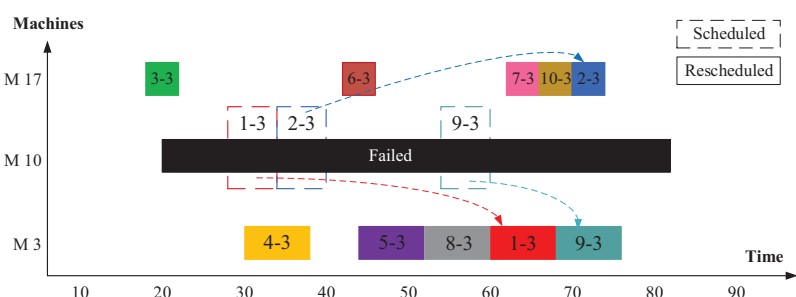

**Figure 8 Rescheduled disrupted operations in Scenario 2.**

For a given processing time $P_{jik}$ for the rescheduled operation, the new end time $NETO_{j(i)kd}$ is found by Eq. (14).

$$NETO_{j(i)kd} = NSTO_{j(i)kd} + P_{jik} \tag{14}$$

where $NETO_{j(i)kd}$ is the new end time of the disrupted operation $O_{j(i)}$, and $P_{jik}$ is the processing time of that operation on machine $k$.

According to the last step, the best executive selected machine is the machine with the earliest end time following Eq. (15).

$$BMO_{j(i)d} = Min\left[NETO_{j(i)dk}\right] \forall \, k \in \{AM_i\} \text{ and } k \notin \{FM\} \tag{15}$$

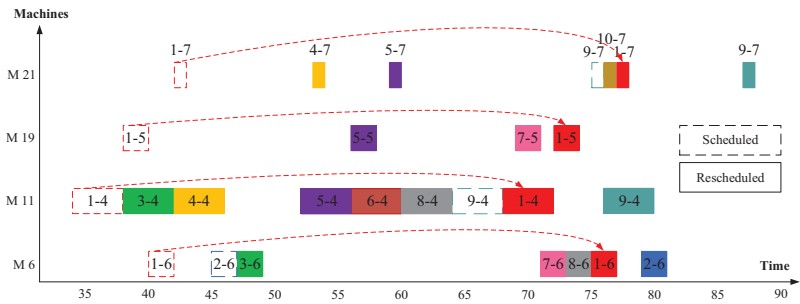

**Figure 9 Rescheduled for the following operations in Scenario 2.**

**Table 4 Disrupted operations in M10 and following operations.**

Disrupted operation → Following operations

$$[1-3] \rightarrow \begin{bmatrix} 1-4 \\ 1-5 \\ 1-6 \\ 1-7 \end{bmatrix} \qquad [2-3] \rightarrow \begin{bmatrix} 2-4 \\ 2-5 \\ 2-6 \\ 2-7 \end{bmatrix} \qquad [9-3] \rightarrow \begin{bmatrix} 9-4 \\ 9-5 \\ 9-6 \\ 9-7 \end{bmatrix}$$

where $BMO_{j(i)d}$ is the selected machine to process the disrupted operation $O_{j(i)}$. $[NETO_{j(i)dk}]$ is the list of new end times for this operation calculated for each alternative machine. $AM_i$ is the set of alternative machines for operation $i$, and $FM$ is the set of failed machines.

Then, the machine availability of the selected machine is updated accordingly.

Second, an explanation of the following operations is configured in Fig. 9 to clarify the rescheduling process of the following operation. Table 4 shows the disrupted operations in M10 and their following operations according to Fig. 2. In this case, the new start and end times are calculated from Eqs. (13) to (15). For the same machine, the machine's availability is then adjusted according to the new end time (Eq. (14)). As a result, we examine the reassign of the following operations of the disrupted operation op(1-3) which are op(1-4), op(1-5), op(1-6) and op(1-7). As shown in Fig. 9, all the following operations are assigned to the same machine that was assigned to before, but there is an extension in the start time of the following operation and the start time can be calculated as in Eq. (12). We will explain a numerical example for op(1-4). The machine M11 is available at the time of 64 after op(8-4) whereas its preceding operation of op(1-3) ends at time of 68. Therefore, operation op(1-4) has been rescheduled at the time of 68. According to Eq. (12), the operation op(1-4) starts at a time of 68. Operation op(1-5) starts as soon as operation op(1-4) ends at the time of 72. For operation op(1-6), the condition of machine availability is the dominant. So, this operation starts after operation op(8-6) ends at the time of 75. Operation op(1-7) will start instantaneously after operation op(1-6) ends, at time 77.

## Energy consumption calculation

To validate the algorithm for calculating energy consumption, we utilized a dataset from *IEEE (2020)*, which contains measurements from a Brazilian poultry feed factory. This

dataset includes readings of active power, reactive power, apparent power, current, and voltage for milling machines, pelletizers, and other factory equipment. The algorithm uses the optimization algorithm results as input to compute the total energy consumption of all machines in the optimal solution. For this evaluation, data was specifically considered for three milling machines, three pelletizers, and three exhaust fans (coolers). Energy consumption was calculated for all production lines, assuming that other machines of the same type (*e.g.*, mixing, liquid addition, scaling, and sewing machines) consume the same amount of energy. Additionally, standby energy consumption for idle machines was disregarded.

For a given dataset of nine machines across three production lines, the total energy consumption of the factory is calculated through the following steps:

- Calculate the average power $P_{avk}$ for each machine over one working day. The average power is derived from the machine's measured power as in Eq. (16).

$$P_t = VI_t \cos\varphi \tag{16}$$

where $P_t$ is the power at time t, $V$ is the voltage on the machine, $I_t$ is the current withdrawn at time t, and $\cos\varphi$ is the power factor.

- Determine the current schedule and count the running time of each machine $RT_k$.
- Compute individual machine energy consumption $EC_k$ by multiplying each machine's average power by its running time according to Eq. (17).

$$EC_k = P_{avk} \times RT_k. \tag{17}$$

- Calculate total energy consumption *TEC* by summing up the energy consumption of all machines.

$$TEC = \sum EC_k \; where \; k \in (1, 4, 5, 8, 11, 12, 15, 18, 19). \tag{18}$$

## SIMULATIONS AND ANALYSIS

The parameter values for the GWO and PSO have been presented in Table 5. All parameters were selected based on standard practices in literature and preliminary experimentation to ensure reasonable performance.

### Generalizability of the proposed GWO and PSO algorithms

Before applying the proposed GWO and PSO algorithms to the poultry factory dataset, their validity and effectiveness have been first evaluated using a well-established benchmark. Specifically, the Brandimarte dataset (BRdata), originally introduced in *Brandimarte (1993)*, is employed. This benchmark comprises ten standard test problems, labeled MK01 through MK10, representing a diverse range of FJSSP scenarios. These instances vary in complexity, with the number of jobs ranging from 10 to 20, machines from four to 15, and operations per job ranging from five to 15. The performance of the proposed algorithms is evaluated over 30 independent computational runs. The numerical results of the proposed GWO and PSO approaches are listed in Table 6 with a comparison

**Table 5 Optimization algorithms parameters.**

| PSO parameter | Value | GWO parameter | Range/Value |
|---|---|---|---|
| Number of iterations | 100 | Number of iterations | 100 |
| Population size (number of particles) | 15 | Population size (Number of Wolves) | 100 |
| Inertia weight | 0.7 | Convergence parameter (a) | Decrease [2-0] |
| Cognitive coefficient (Personal learning factor) | 1.5 | Distance factor (A) | [−2:2] |
| Social coefficient (Global learning factor) | 1.5 | Distance factor (C) | Random in [0, 2] |

**Table 6 Results in terms of makespan of the Brandimarte instances for different algorithms.**

| Test case | n × m | GA by (*Naimi, Nouiri & Cardin, 2021*) | Hybrid GA by (*Sun et al., 2023*) | SS-GWO By (*Zhou et al., 2024*) | LHS by (*Li & Zhou, 2025*) | Proposed PSO | | | Proposed GWO | | |
|---|---|---|---|---|---|---|---|---|---|---|---|
| | | | | | | Makespan | Average makespan | Average execution time (seconds) | Makespan | Average makespan | Average execution time (seconds) |
| Mk01 | 10 × 6 | 42 | 40 | 39 | 40 | 41 | 42.27 | 14.98 | 40 | 41.9 | 7.77 |
| MK02 | 10 × 6 | 32 | 26 | 36 | 26 | 28 | 28.36 | 16.46 | 27 | 28.1 | 11.6 |
| MK03 | 15 × 8 | 206 | 204 | 228 | 204 | 204 | 204 | 50.58 | 204 | 204 | 37.21 |
| MK04 | 15 × 8 | 67 | 60 | 72 | 60 | 66 | 66.9 | 17.43 | 64 | 66.9 | 13.29 |
| MK05 | 15 × 4 | 179 | 173 | 179 | 171 | 177 | 179.9 | 30.33 | 173 | 177.1 | 21.99 |
| MK06 | 10 × 15 | 86 | 61 | 84 | 55 | 65 | 66.96 | 54.21 | 63 | 66.2 | 36.83 |
| MK07 | 20 × 5 | 164 | 140 | 173 | 141 | 146 | 149.03 | 32.11 | 143 | 146.13 | 24.19 |
| MK08 | 20 × 10 | 523 | 523 | 541 | 523 | 523 | 523 | 74.69 | 523 | 523 | 42.57 |
| MK09 | 20 × 10 | 342 | 307 | 378 | 301 | 313 | 316.76 | 105.91 | 307 | 313.76 | 63.46 |
| MK10 | 20 × 15 | 292 | 214 | 282 | 205 | 231 | 234.36 | 112 | 221 | 227.4 | 67.04 |

to the outcomes from prior studies (*Zhou et al., 2024*) and (*Naimi, Nouiri & Cardin, 2021*; *Sun et al., 2023*; *Li & Zhou, 2025*) that utilized alternative optimization techniques to assess the competitiveness and generalizability.

The experimental results highlight the effectiveness of both PSO and GWO algorithms in solving the Brandimarte benchmark set MK01–MK10 (in Table 6). Across the ten test cases, both approaches deliver competitive makespans that align closely with, and in some instances match, the best-known results reported in recent literature. For example, in MK01 and MK03, both PSO and GWO achieve makespans near or equal to the global optima, demonstrating their reliability on standard instances. In more complex problems such as MK06, MK09, and MK10, the algorithms maintain strong performance, producing makespans within a narrow margin of the best reference values in *Li & Zhou (2025)*. While PSO often requires slightly more execution time, it consistently delivers quality results across problem sizes. The GWO, on the other hand, shows a tendency for faster convergence in some cases. Overall, both algorithms can handle a wide range of flexible job shop configurations, validating the proposed algorithm's adaptability and robustness in addressing real-world scheduling challenges.

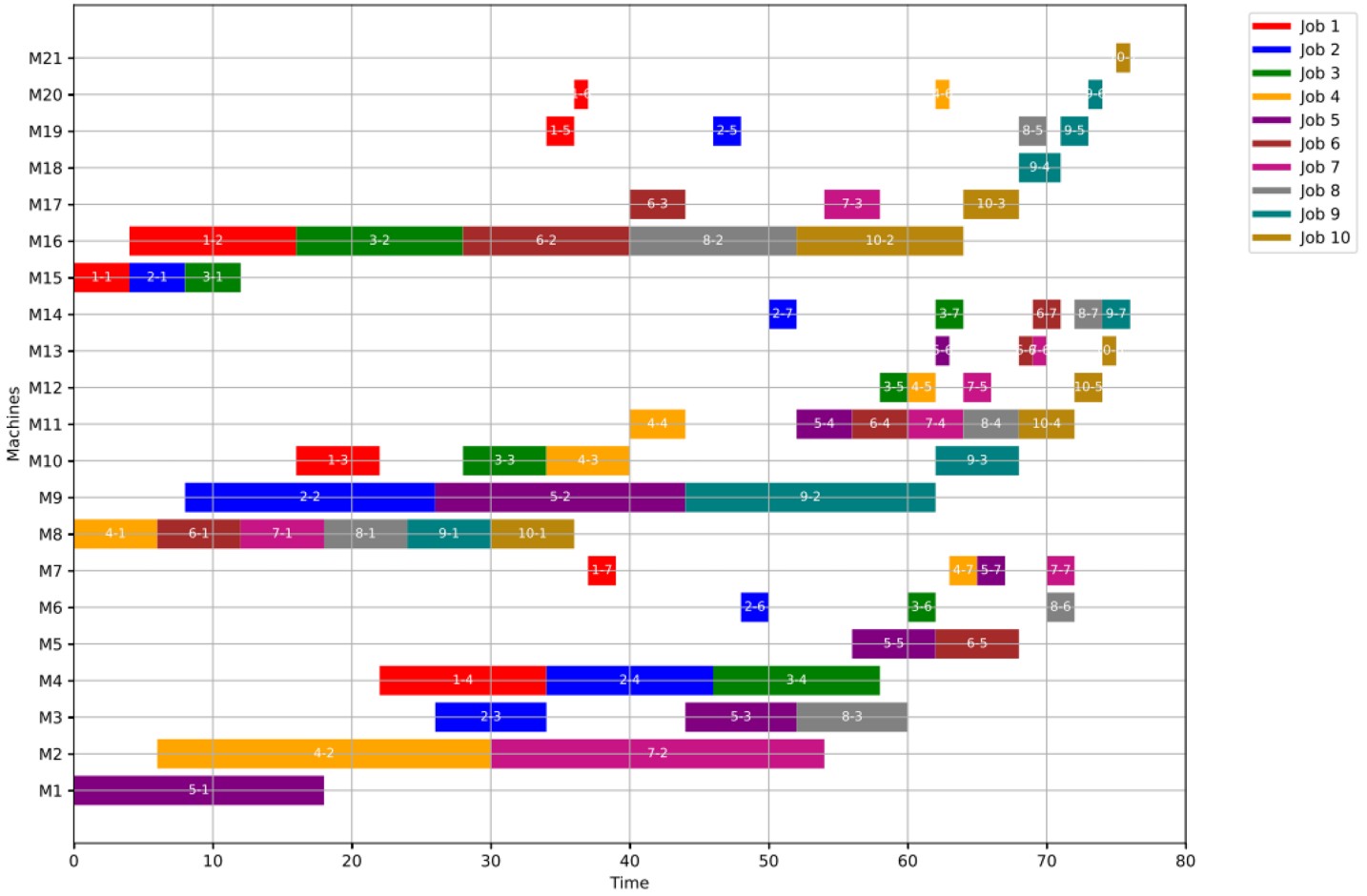

**Figure 10  Gantt chart for healthy state using GWO with makespan = 76 time units.**

## Generating the healthy scheduling plan

Generating the healthy scheduling plan is the foundational step in job-shop scheduling, as it represents the optimal sequence and machine assignments for all operations under normal operating conditions, *i.e.*, without any disturbances such as machine failures. In this stage, metaheuristic optimization algorithms including both GWO and PSO are employed to explore the solution space and determine a schedule that minimizes the makespan while satisfying all precedence and resource constraints. The healthy schedule not only ensures that operations are allocated to machines in a manner that avoids conflicts and idle times but also forms the baseline against which any rescheduling strategies are evaluated in the event of disruptions. It reflects the best-case scenario in terms of efficiency and serves as a reference for assessing the impact of failure scenarios on production performance.

For GWO, the healthy schedule is generated as shown in Fig. 10 with minimum makespan = 76, achieved at the 44[th] iteration. While For PSO, the healthy schedule is generated as shown in Fig. 11 with minimum makespan = 76, achieved at the 47[th] iteration.

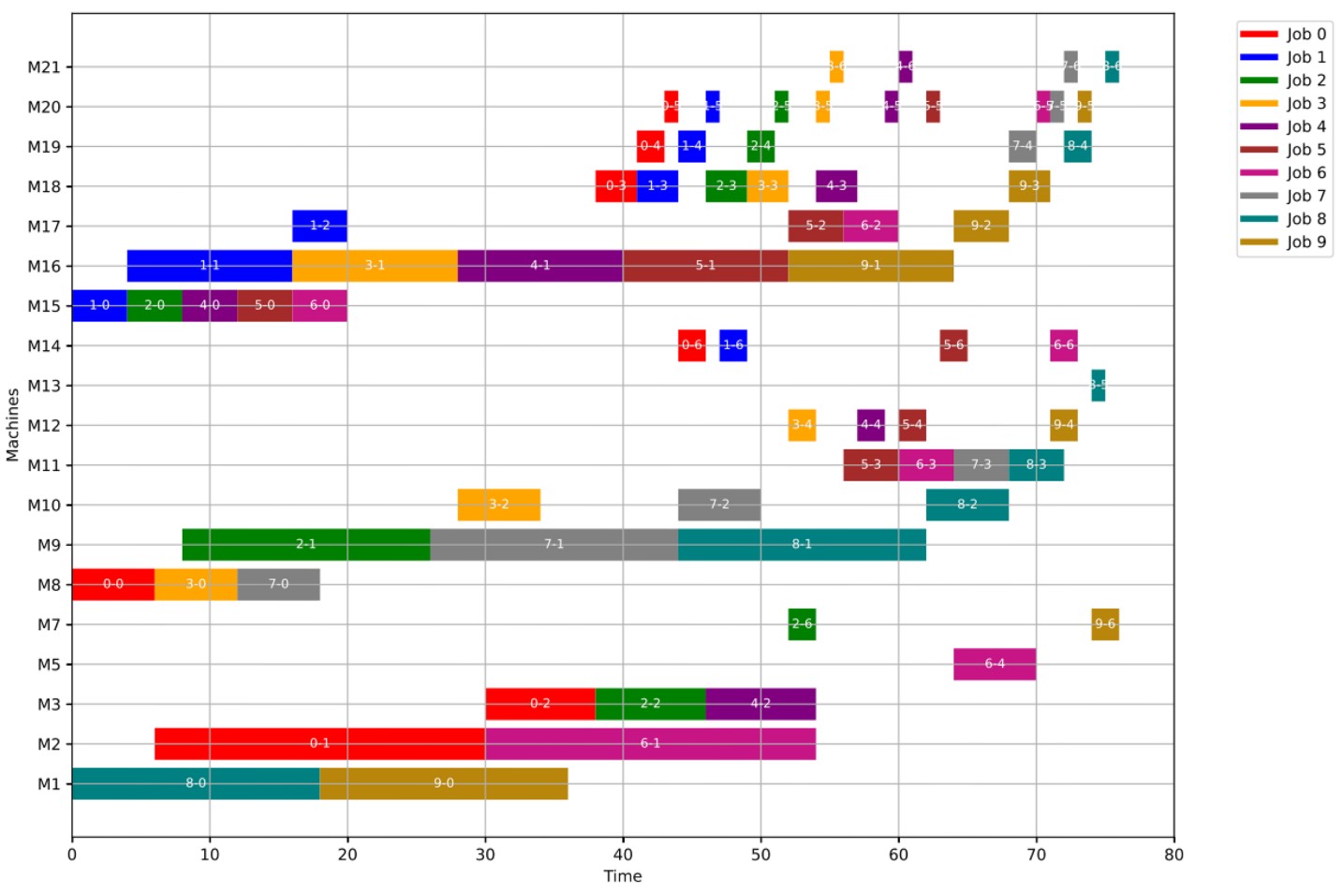

**Figure 11 Gantt chart for healthy state using PSO with makespan = 76 time units.**

From Figs. 10 and 11, it can be noticed that both algorithms achieved the same minimum makespan. The difference is in the distribution of operations on the machines.

## Optimizing the completion time

In this section, we present a comprehensive numerical analysis to evaluate the performance of our proposed algorithm across several examples. For each case study, the analysis begins with the evaluation of the healthy schedule, which is generated using either GWO or PSO. Since both algorithms yield identical makespan values in the healthy state, the distinction lies solely in the distribution of operations across the machines. Subsequently, we examine the results under Scenario 1 using both GWO and PSO. While the outcomes are generally consistent, minor variations may occur due to algorithmic differences. Lastly, we assess Scenario 2, which does not involve any optimization algorithm but instead relies solely on a shifting mechanism to reschedule operations.

Three case studies are explored to demonstrate the impact of key parameters—namely, the failure time and the identity of the failed machines—on scheduling performance. These case studies were carefully selected to highlight the influence of these primary factors. As

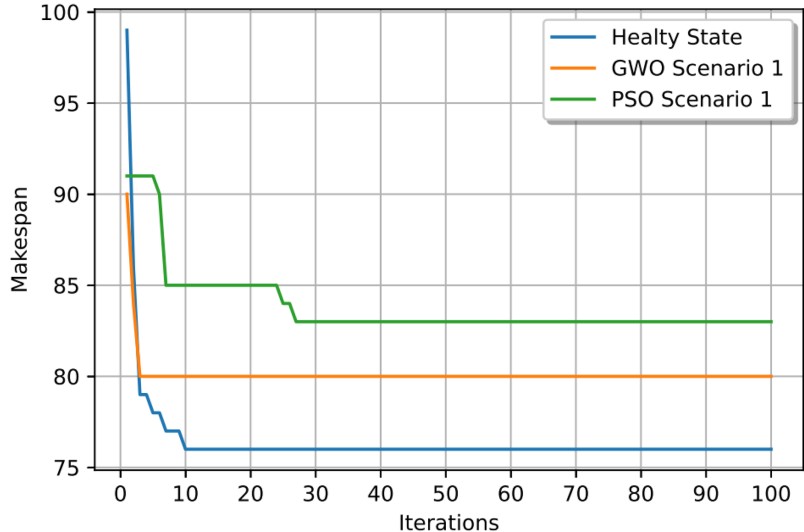

**Figure 12 Convergence curves: fitness function for: the healthy state before failure, GWO and PSO after the failure for Scenario 1.**

expected, failures occurring later in the schedule lead to smaller increases in makespan, since a larger portion of operations have already been completed, leaving fewer to be rescheduled. Additionally, the criticality of the failed machine significantly affects the results: failures in major machines, such as milling or pelletizing units, have a more substantial impact on the scheduling outcome compared to failures in less critical equipment like sewing machines, packing units, or coolers.

In the first case study, the failure occurs at time unit 20, affecting two critical machines: M1 and M10. In the second case study, although the failure time remains the same at 20, the failed machines are different—M8 and M18—which alters the rescheduling dynamics and potentially reduces the impact on the overall schedule depending on the roles of these machines. The third case study maintains the same failed machines as the second one (M8 and M18), but shifts the failure time to 30. This adjustment allows more operations to be completed before the disruption, leading to a less pronounced effect on the makespan.

These variations across the case studies were purposefully designed to demonstrate the sensitivity of the schedule to both the timing and the location of machine failures. The next section provides a focused analysis of energy consumption using these same three case studies, allowing for a comparative evaluation not only of scheduling efficiency but also of operational sustainability.

**A) Case study (1)**

In this case study, the failure occurs at time 20, affecting machines M1 and M10. Prior to the anomaly, the optimal makespan is 76, achieved at the 10th iteration. Following the failure event, both the GWO and PSO algorithms optimally redistribute operations among the remaining healthy machines. Under Scenario 1, the optimal makespan after the failure is 80 when using GWO and 83 when using PSO. The convergence curves are presented in Fig. 12. It is notable that the fitness curves after the anomaly are nearly horizontal, as the

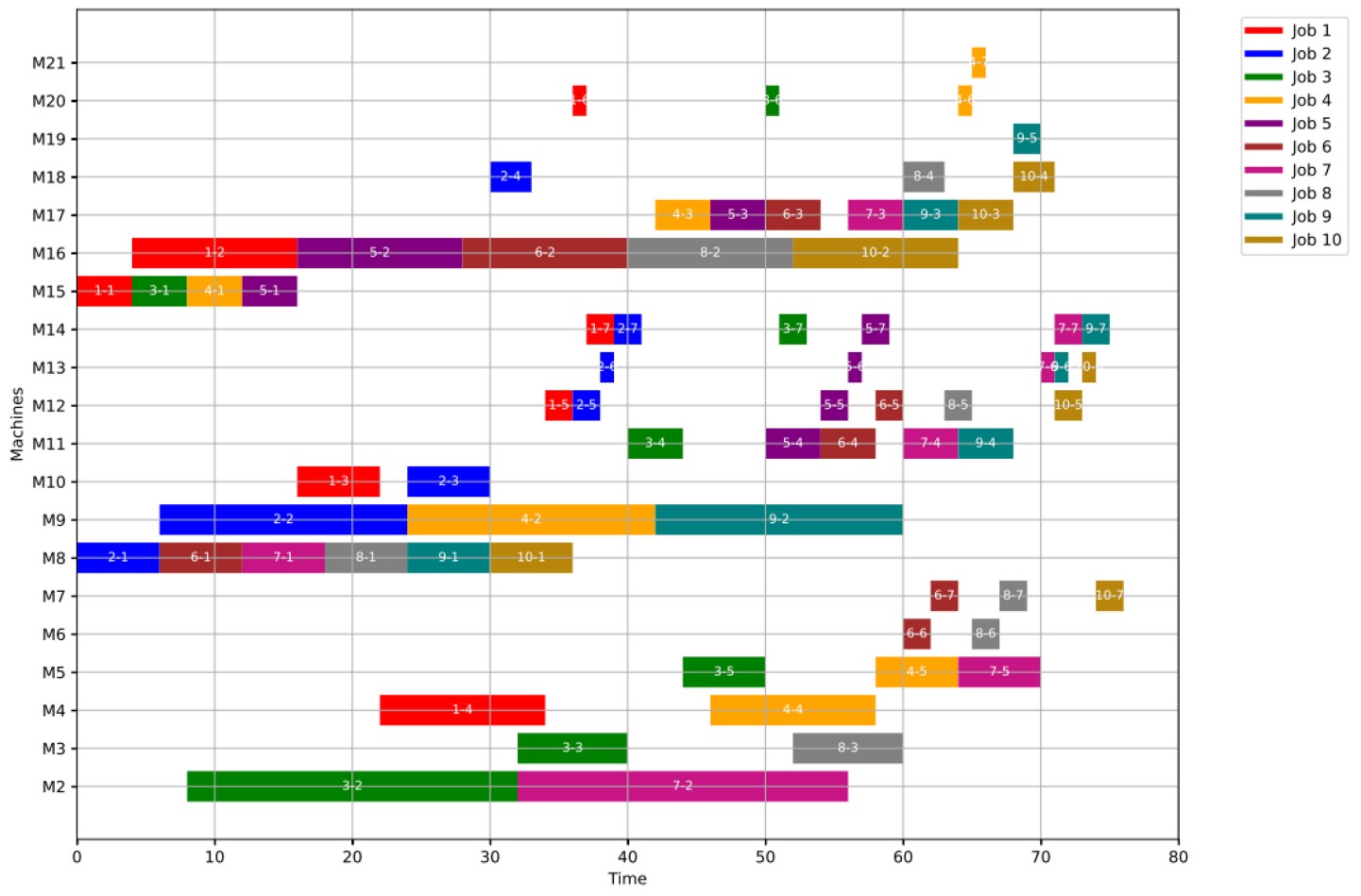

**Figure 13 Initial gantt chart for healthy state: case study (1).**

optimization process is initialized using the pre-anomaly optimal schedule, allowing for faster convergence.

Figures 13, 14, and 15 illustrate the healthy operation case, the GWO-optimized schedule under Scenario 1, and the PSO-optimized schedule under Scenario 1, respectively. Job operations are allocated across machines based on the results of the optimization algorithm presented in Fig. 10. A dynamic event (machine failure) is detected at the failure time $T_F = 20$, triggering the rescheduling process.

Machines M1 and M10 are identified as having anomalous operations. In this scenario, all the completed operations remain unchanged, while the operations ongoing at $T_F = 20$ are treated in two distinct ways. For the healthy machines, the operations in progress at $T_F = 20$ are termed executing operations and these operations remain intact, and they are still assigned to their initial machines in order not to experience a delay in the makespan completion time. For the failed machines at $T_F = 20$, the operations in progress are termed disrupted operations. Furthermore, operations scheduled after $T_F = 20$ are all categorized as remaining operations along with the disrupted operations. In this case, remaining operations are redistributed in accordance with the adopted evolutionary optimization

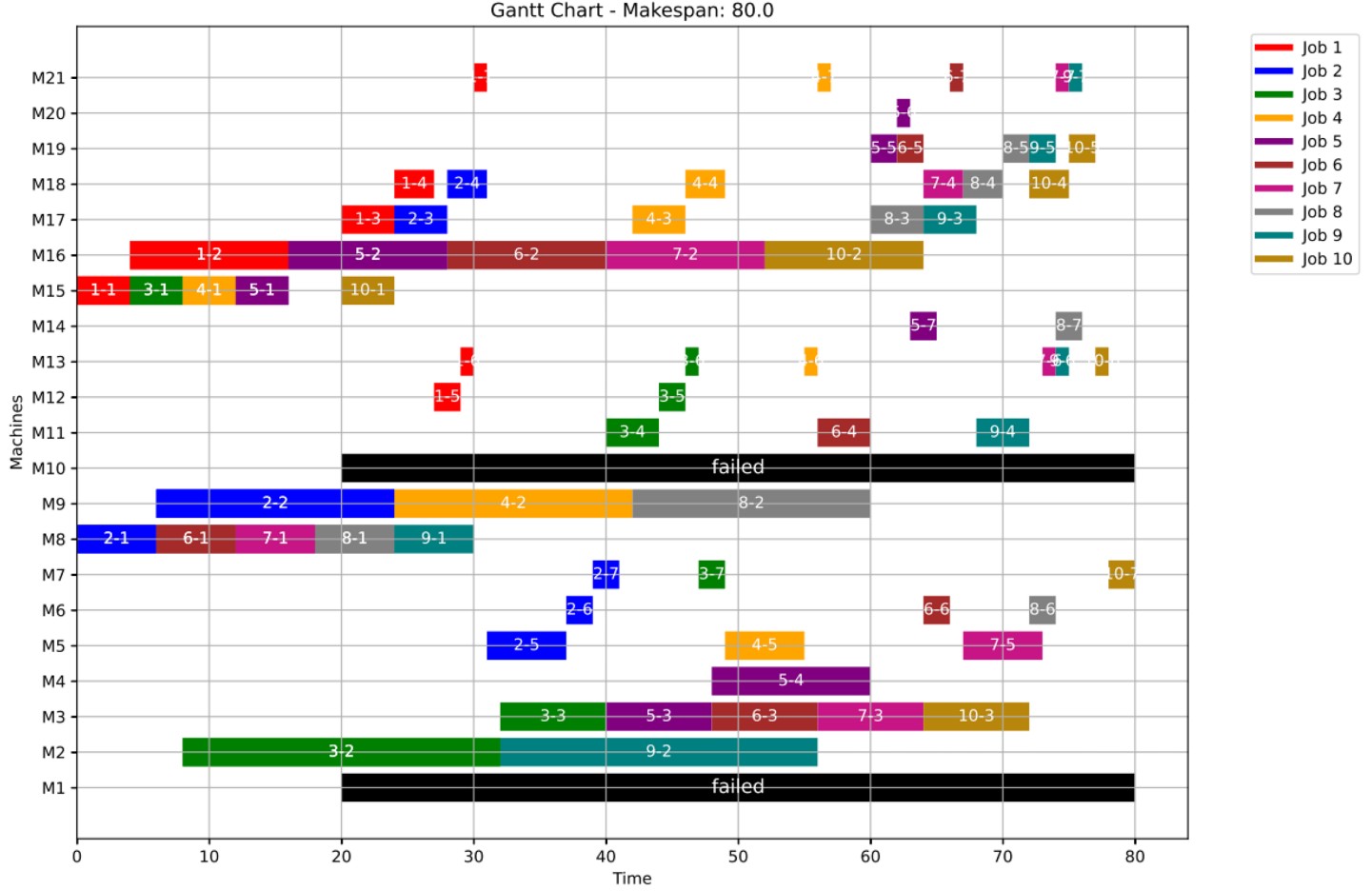

**Figure 14 Gantt chart applying GWO with failure in M1 and M10 at disrupted time $T_F$ = 20: case study (1) of Scenario 1.**

algorithm. At the failure time $T_F = 20$, all remaining operations are redistributed using the proposed GWO approach, based on Eqs. (5)–(8) in 'Initial Scheduling', and the proposed PSO approach, based on Eqs. (9) and (10). It is important to note that the time required to rerun the optimization algorithms is not included in the makespan calculation.

Figure 16 shows the Gantt chart when applying Scenario 2. The makespan in this case is 87, the disrupted operations are op(1-3) and op(2-3). Op(1-3) is reassigned to M3 and op(2-3) is reassigned to M17. Consequently ops(1-4:7) and ops(2-4:7) are extended to be processed on the same assigned machine as the healthy case but shifted in time until the previous operation is done or the machine is available.

As a numerical case study for Scenario 1, Job 2 (illustrated in blue in Fig. 11) originally follows a sequence of seven operations processed on seven different machines: M8, M9, M10, M18, M12, M13, and M14. This operation sequence is determined in the healthy state based on the predefined machine selection and scheduling plan. Specifically, op(2-1) begins execution on M8 and is immediately followed by op(2-2) on M9, and then op(2-3) on M10, reflecting a continuous and optimized workflow.

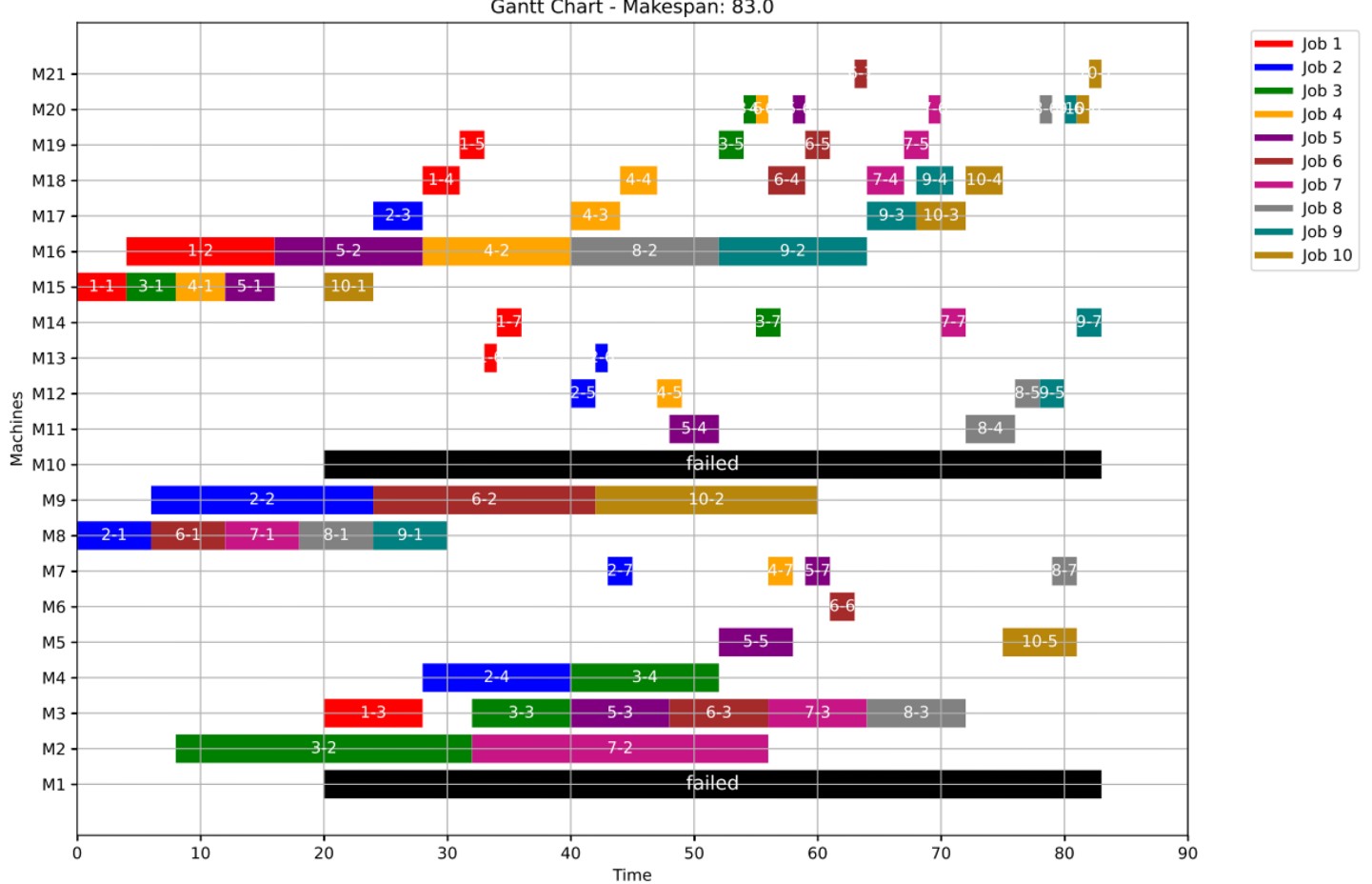

**Figure 15 Gantt chart applying PSO with failure in M1 and M10 at disrupted time $T_F = 20$: case study (1) of Scenario 1.**

Upon the occurrence of a machine failure, ops(2-1) and (2-2) are left unchanged—(2-1) is already completed, and (2-2) is in execution—thus, both are preserved in the new schedule as finished and executing operations, respectively. However, op(2-3), which was originally scheduled on the now-failed machine M10, is reassigned to an alternative machine, M17. The remaining operations of Job 2 are subsequently redistributed while maintaining their original order, adapting to the updated resource availability.

This rescheduling process successfully preserves the consistency of the job's execution and minimizes delays in the overall completion time. It demonstrates the robustness and adaptability of the scheduling strategy under disruption, ensuring continuity and efficiency even in the presence of machine failures.

**B) Case study (2)**

In this case study, the failure time equals 20 and the failed machines are M8 and M18. The optimal makespan, prior to the anomaly, is 76 at the 46th iteration. However, once the anomaly event occurred, the GWO and the PSO algorithms optimally redistribute the operations among all healthy machines. After the failure and when running Scenario 1, the

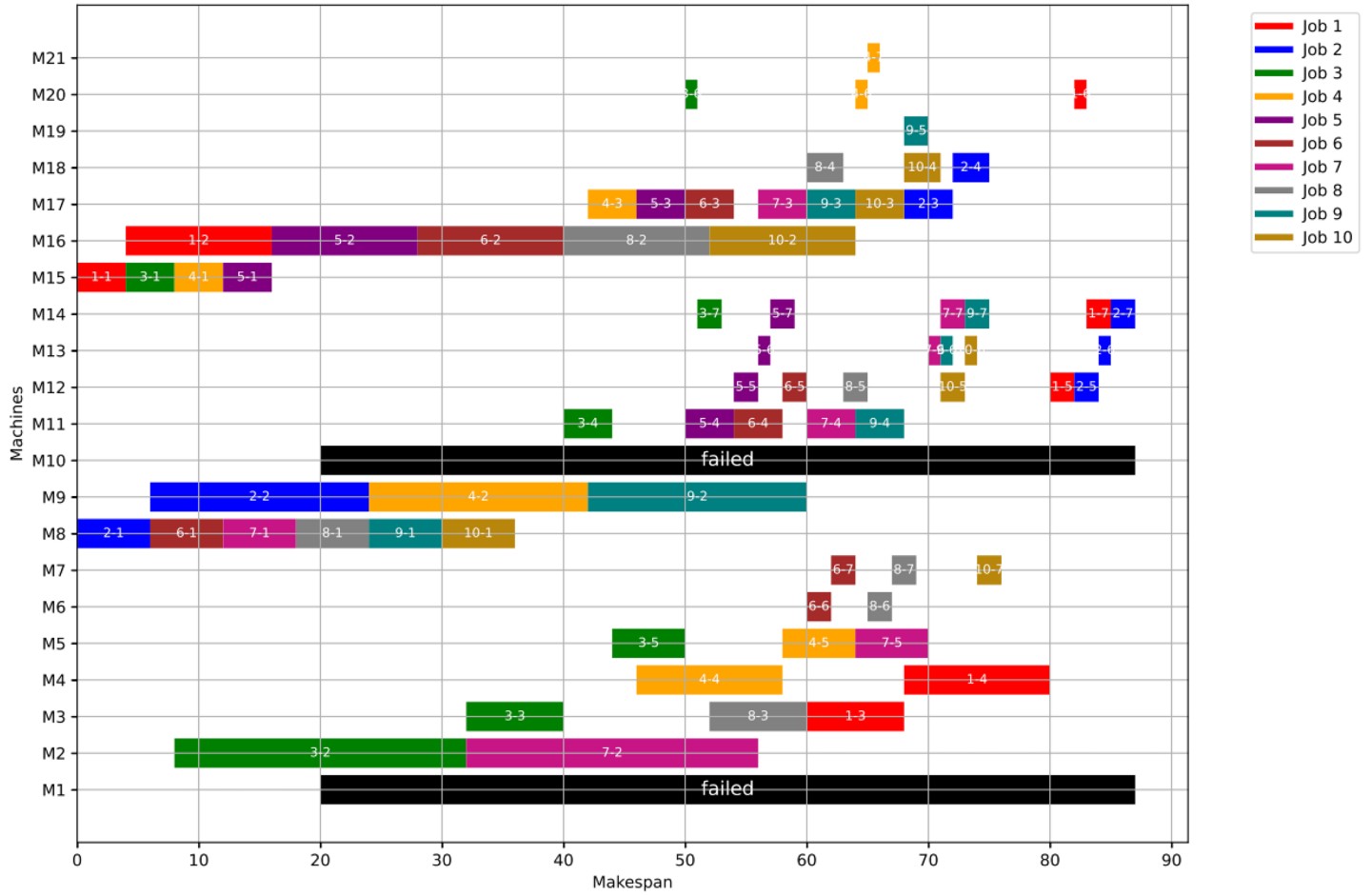

**Figure 16 Gantt chart with failure in M1 and M10 at disrupted time $T_F$ = 20: case study (1) of Scenario 2.**

optimal makespan is 85 while with the PSO is also 86. The convergence curves are shown in Fig. 17.

Figures 18, 19, and 20 illustrate the healthy operation case, the GWO-optimized schedule under Scenario 1, and the PSO-optimized schedule under Scenario 1, respectively, each depicting the system's behaviour under anomaly conditions. While job operations are allocated among the machines based on the results of the optimization algorithm shown in Fig. 10, a dynamic event has been detected at the failure time of $T_F = 20$. Machines M8 and M18 are identified as having anomalous operations. In this scenario, all the completed operations remain unchanged, while the operations ongoing at $T_F = 20$ are treated in two distinct ways. For the healthy machines, the operations in progress at $T_F = 20$ are termed executing operations and these operations remain intact, and they are still assigned to their initial machines in order not to experience a delay in the makespan completion time. For the failed machines at $T_F = 20$, the operations in progress are termed disrupted operations. Furthermore, operations scheduled after $T_F = 20$ are all categorized as remaining operations along with the disrupted operations. In this case,

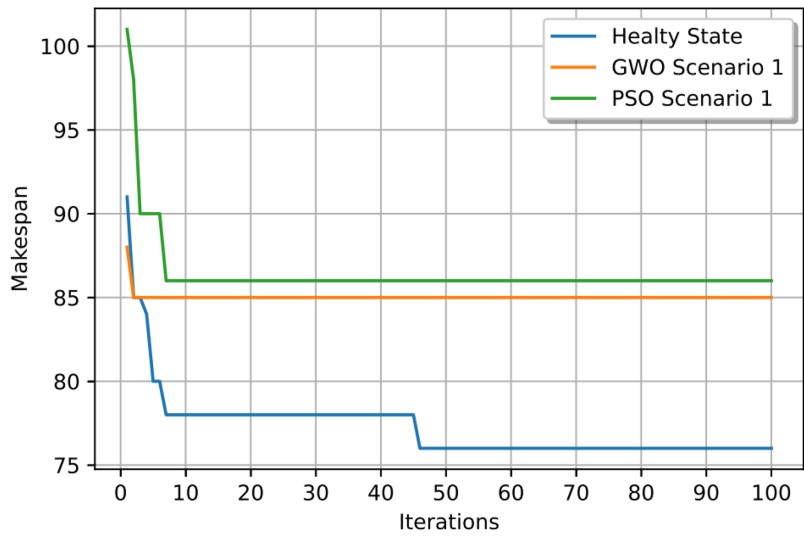

**Figure 17 Convergence curves: fitness function for: the healthy state before failure, GWO and PSO after the failure: case study (2) of Scenario 1.**

**Figure 18 Initial Gantt chart for healthy state: case study (2).**

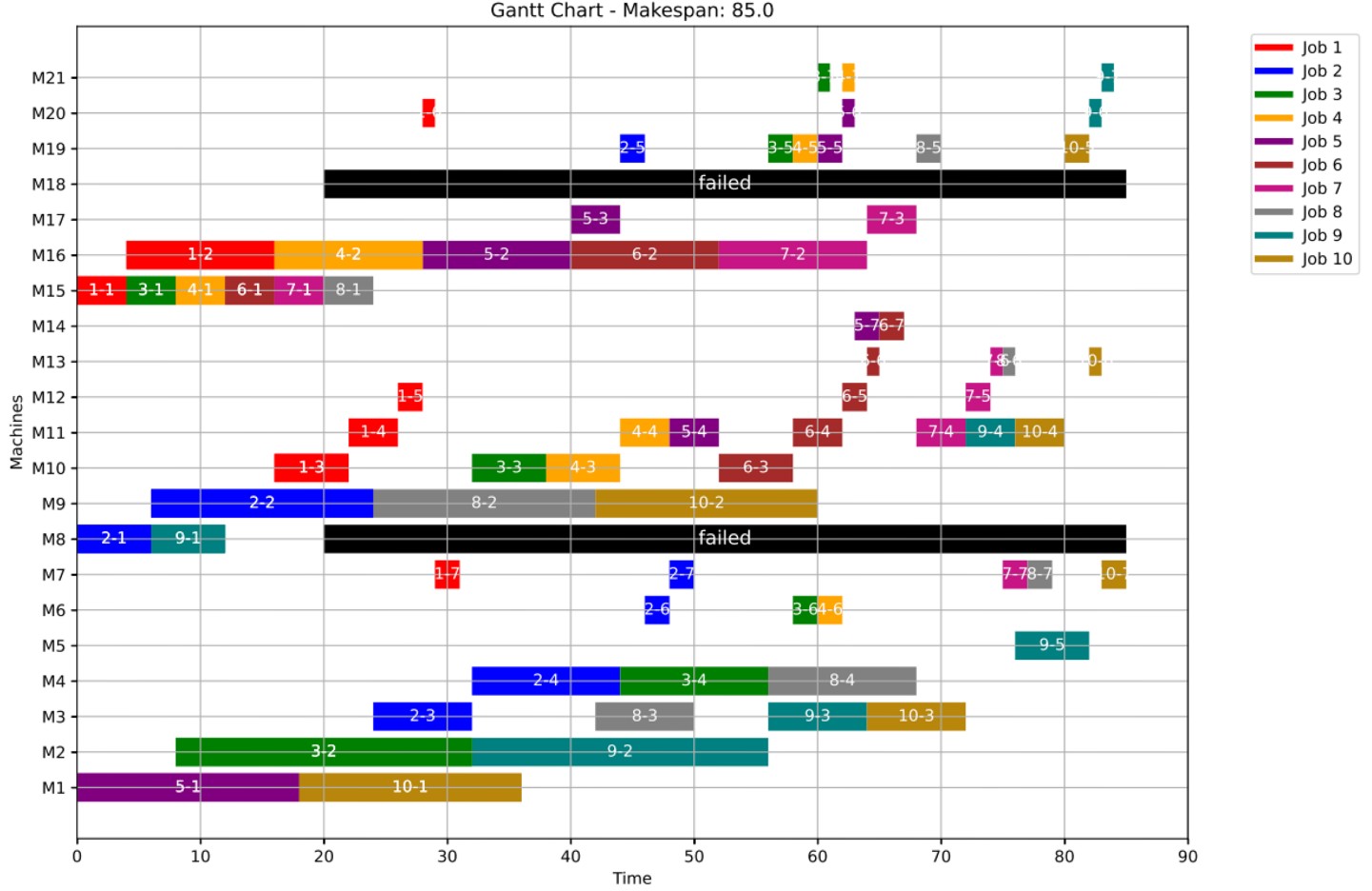

**Figure 19 Gantt chart applying GWO with failure in M8 and M18 at disrupted time $T_F$ = 20: case study (2) of Scenario 1.**

remaining operations are redistributed in accordance with the adopted evolutionary optimization algorithm.

At the failure time $T_F = 20$, all remaining operations are redistributed using the proposed GWO approach, as defined by Eqs. (5)–(8), and the proposed PSO approach, as defined by Eqs. (9) and (10), all in 'Initial Scheduling'.

Figure 21 presents the Gantt chart corresponding to Scenario 2. In this case, the makespan reaches 81. The disruption involves two operations: op(7-4) and op(9-4). Due to the failure of their originally assigned machines, op(7-4) is reassigned to machine M4, while op(9-4) is reassigned to machine M11. As a result of these reassignments, the subsequent operations—ops(7-5) through (7-7) and ops(9-5) through (9-7)—maintain their original machine assignments as defined in the healthy schedule. However, these operations are shifted in time, ensuring they start only after their preceding operations are completed and their respective machines become available. This time-shifting strategy preserves operational precedence and resource feasibility, ensuring the continuity of job execution with minimal additional delays.

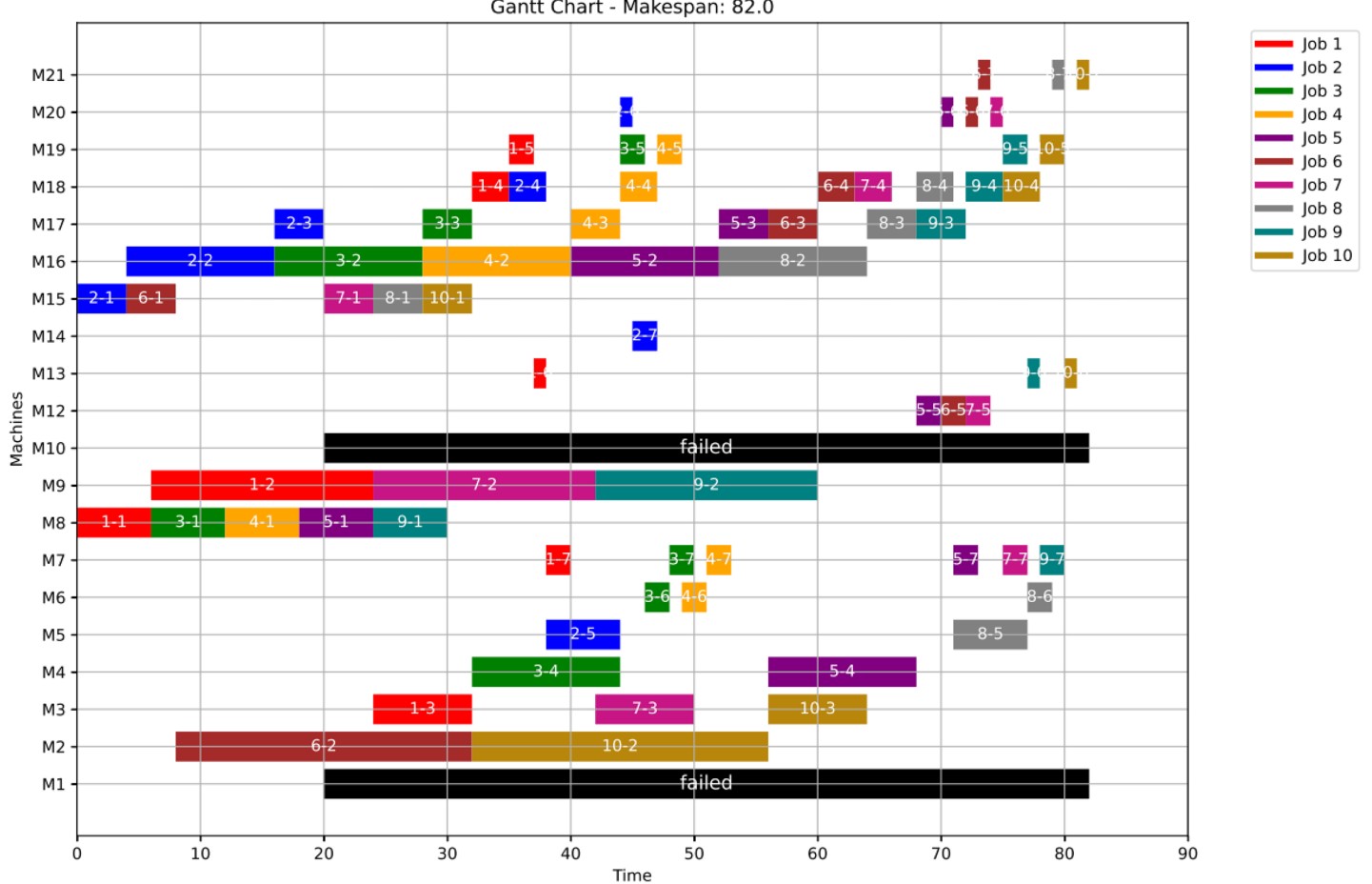

**Figure 20 Gantt chart applying PSO with failure in M8 and M18 at disrupted time $T_F$ = 20: case study (2) of Scenario 1.**

## C) Case study (3)

In this case study, the failure occurs at time 30, affecting machines M8 and M18. Prior to the anomaly, the optimal makespan is 76, achieved at the 32nd iteration. Following the failure event, both the GWO and PSO algorithms optimally redistribute the remaining operations among the healthy machines. Under Scenario 1, the optimal makespan after the failure is 82 when using GWO, and 83 when using PSO. The corresponding convergence curves are presented in Fig. 22.

Figures 23, 24, and 25 illustrate the healthy operation case, the GWO-optimized schedule under Scenario 1, and the PSO-optimized schedule under Scenario 1, respectively, all under anomaly conditions. While job operations are allocated among the machines based on the results of the optimization algorithm shown in Fig. 10, a dynamic event has been detected at the failure time of $T_F = 30$. Machines M8 and M18 are identified as having anomalous operations.

Figure 26 shows the Gantt chart when applying Scenario 2. The makespan in this case is 88, the disrupted operations are op(4-4) ,op(5-4) and op(9-4). Op(4-4) is reassigned to

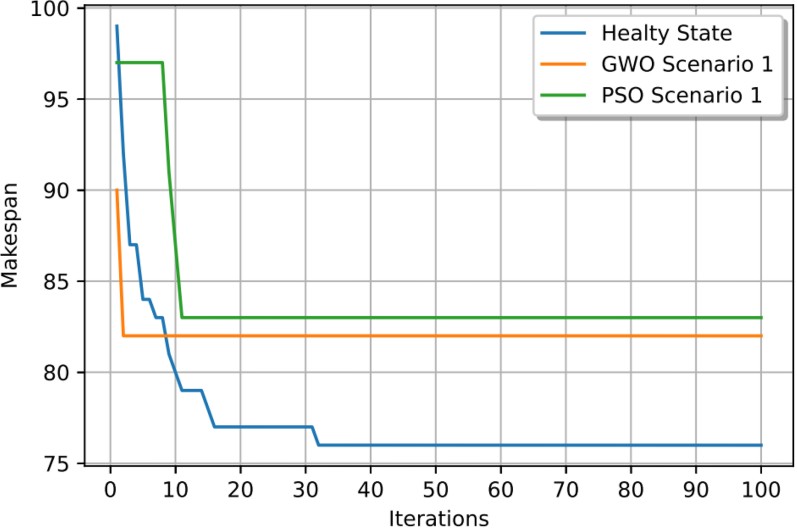

**Figure 21 Gantt chart with failure in M8 and M18 at disrupted time $T_F = 20$: case study (2) of Scenario 2.**

**Figure 22 Convergence curves: fitness function for: the healthy state before failure, GWO and PSO after the failure: case study (3) of Scenario 1.**

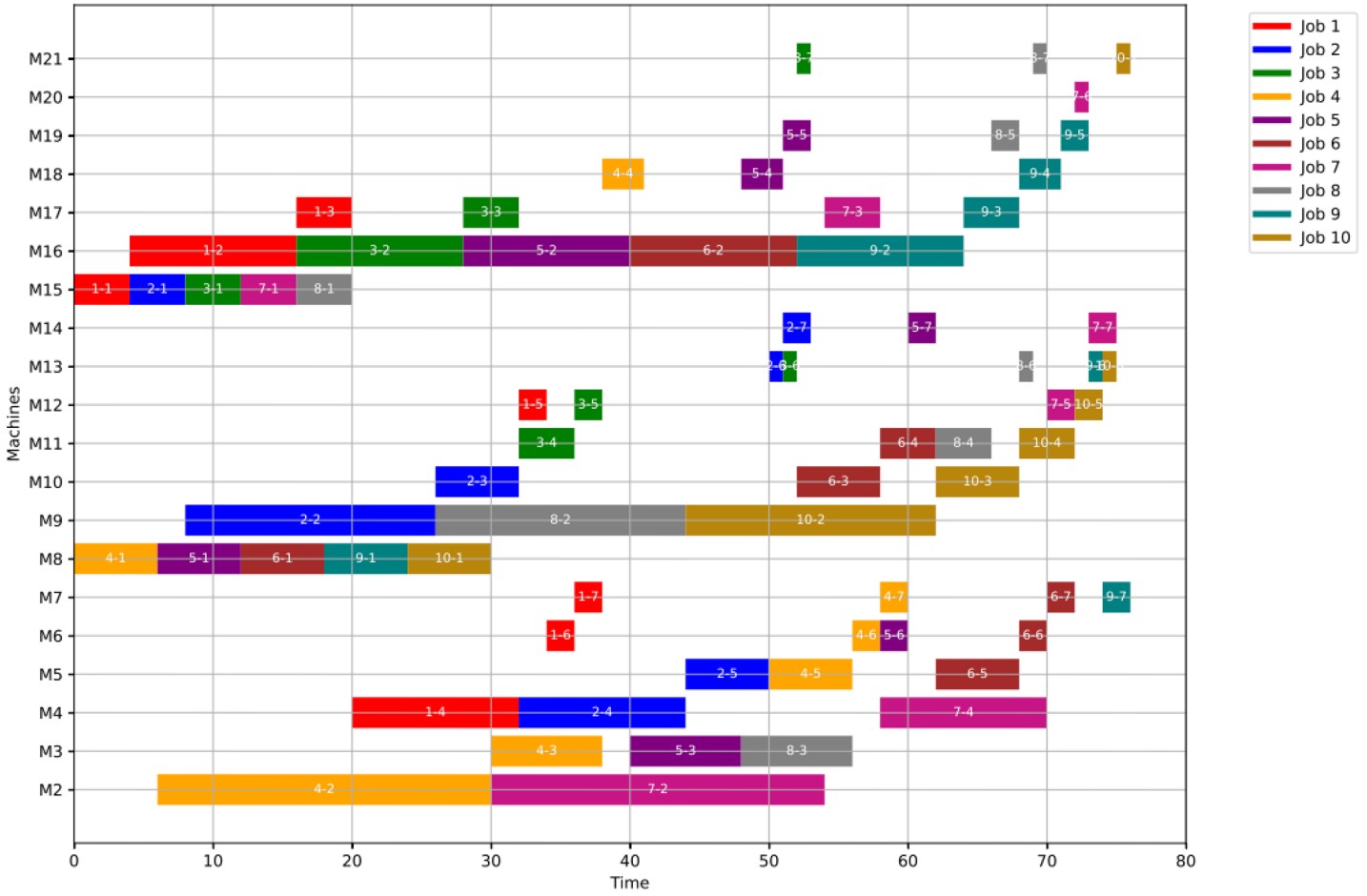

**Figure 23 Initial Gantt chart for healthy state: case study (3).**

M11 and op(5-4) is reassigned to M11, while op(9-4) is reassigned to M4. Consequently ops(4-5:7), ops(5-5:7) and ops(9-5:7) are extended to be processed on the same assigned machine as the healthy case but shifted in time until the previous operation is done or the machine is available.

Table 7 presents the start and completion times for all jobs in the healthy state and after the failure of machines M1 and M10 (case study 1), using both Scenario 1 and Scenario 2. For Scenario 1, results are shown for both GWO and PSO implementations. It is observed that most jobs begin at the same time across the healthy state, Scenario 1, and Scenario 2. However, the key distinction lies in the jobs' completion times. While some jobs experience only minor delays, others are significantly affected, with completion time extensions reaching up to 48 time units.

Table 8 provides a summary of the results from the three case studies, including the healthy state, Scenario 1 using GWO, Scenario 1 using PSO, and Scenario 2.

In Fig. 27, a pairwise comparison is presented, illustrating the extended makespan resulting from simulating failures in each of the alternative machines (one at a time) for three specific operations: Operation 1 (milling) depicted in Fig. 27A, Operation 3

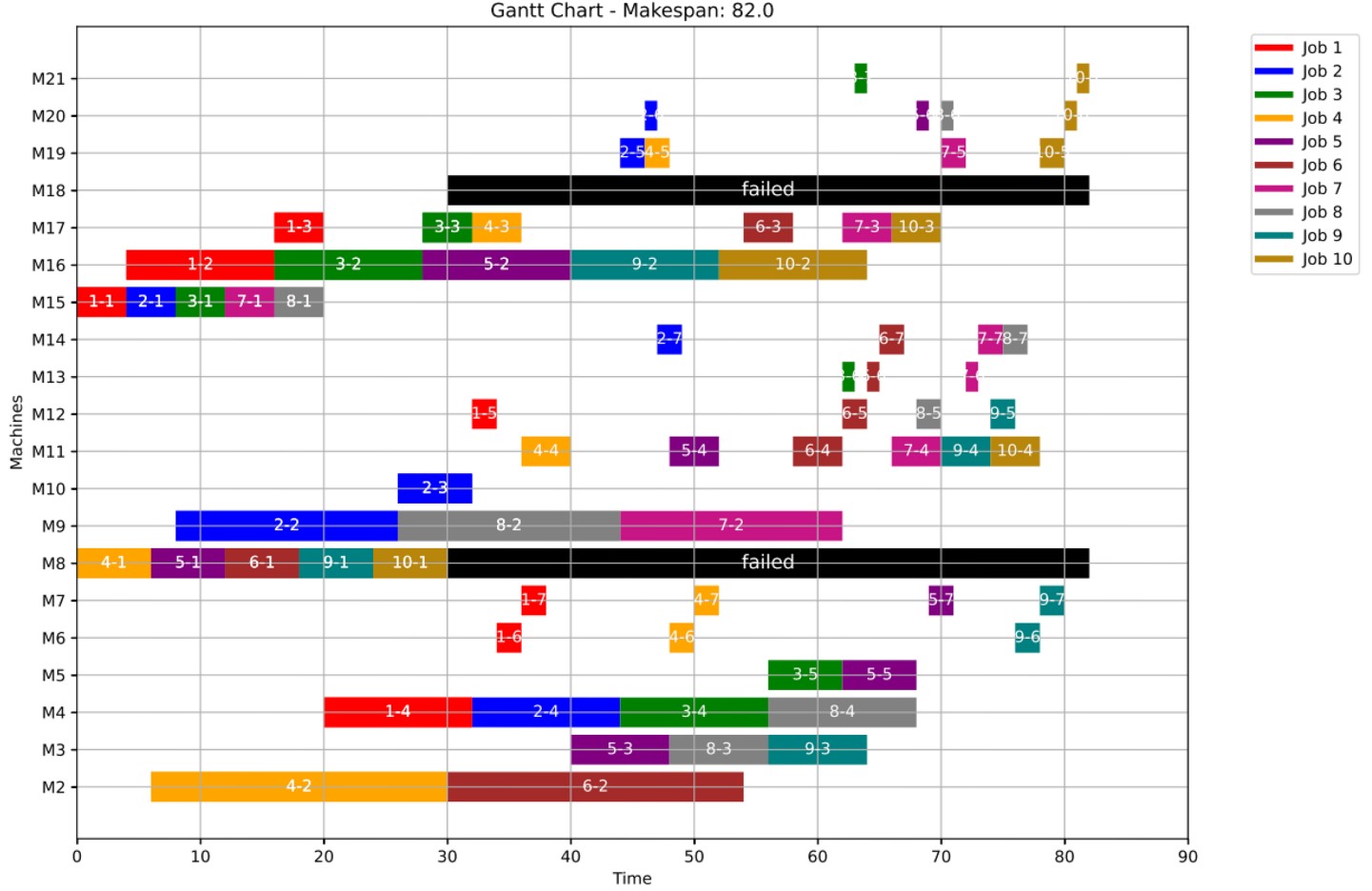

**Figure 24 Gantt chart applying GWO with failure in M8 and M18 at disrupted time $T_F$ = 30: case study (3) of Scenario 1.**

(adding liquid) as shown in Fig. 27B, and Operation 5 (cooling) compared in Fig. 27C. The comparison is done in both Scenario 1 and Scenario 2. For instance, the milling operation can be completed by machine M1 in 18 time units, while machines M8 and M15 can handle the same operation in 6 and 4 time units, respectively. This implies that if a fault occurs in a high-power rating machine like M15, which is assigned to four or five operations, the makespan extension could be significant.

Redistributing these operations to lower-power rating machines would result in considerably longer processing times. But this is not the only factor to be weighed as there is also the impact of the failure time and how busy the alternatives are. When M1 is experiencing failure at time 10, the workload will be distributed upon both M8 and M15. But due to its long processing time (18 units of time), it was not loaded with many operations in the initial schedule. This led to only 12% makespan extension using Scenario 1 and 29% using Scenario 2.

When studying the add liquid operation, M3 can handle the operation in 8 time units, while machines M10 and M17 can handle the same operation in 6 and 4 time units,

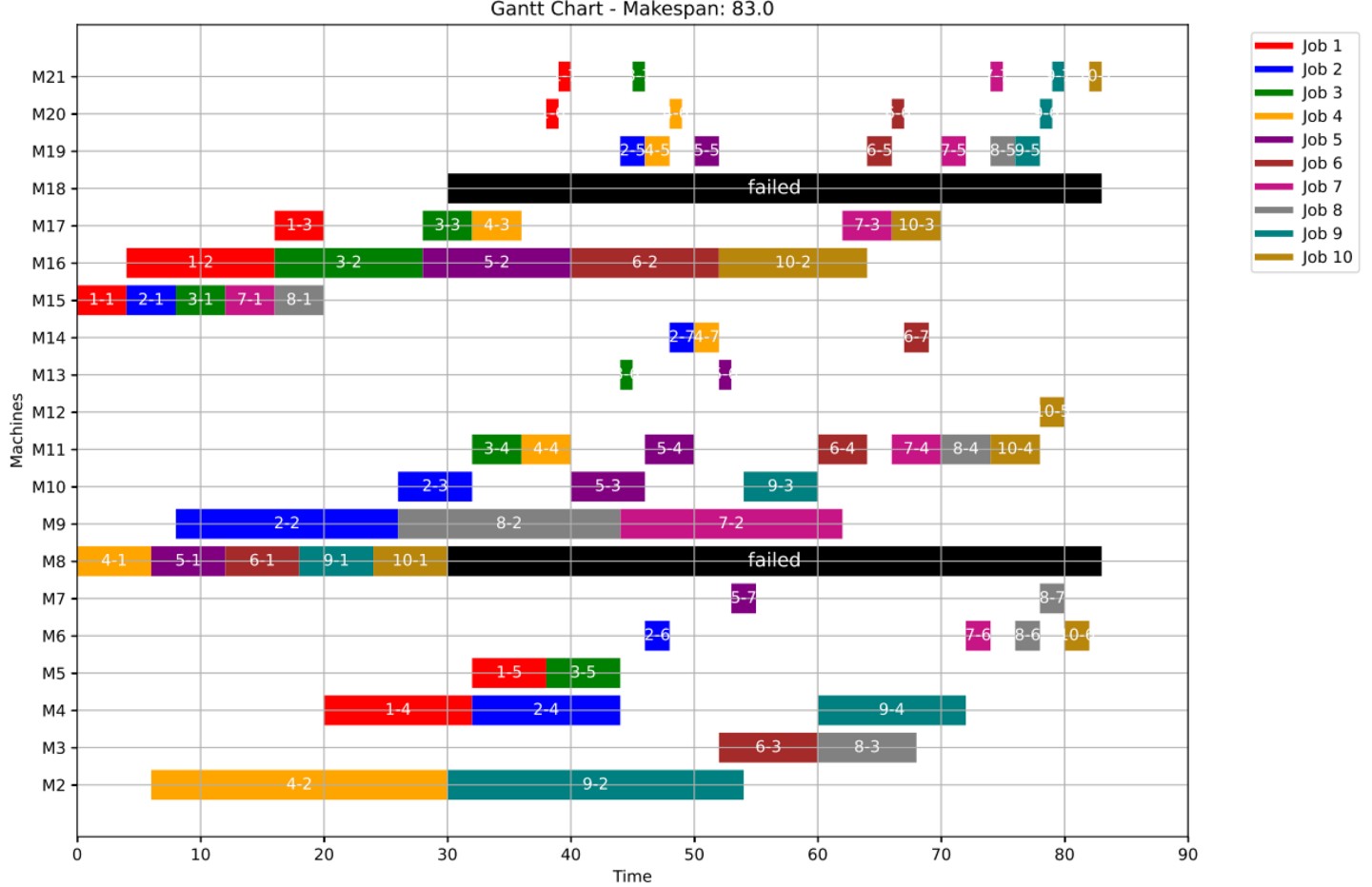

**Figure 25 Gantt chart applying PSO with failure in M8 and M18 at disrupted time $T_F = 30$: case study (3) of Scenario 1.**

respectively. If M10 fails at time 10, its scheduled operations will be distributed between M3 and M17 leading to 10% makespan extension when rescheduling with Scenario 1 and 15% with Scenario 2.

## Energy consumption before and after failure

In this subsection, the estimated energy consumption is recorded for three simulated scenarios for both GWO and PSO optimization algorithms such as: (1) healthy operation with GWO scheduling, (2) faulted operation handled using Scenario 1, and (3) faulted operation handled using Scenario 2. Table 9 presents the estimated energy consumption for these three operational states. As observed, energy consumption in the healthy state is the lowest. When comparing the two fault-handling scenarios, Scenario 2 exhibits lower energy consumption. This is because Scenario 2 applies GWO and PSO optimization only once, followed by a machine break-down shifting technique. In contrast, Scenario 1 applies the GWO and PSO both before and after the anomaly event, redistributing operations to minimize the maximum makespan without explicitly considering energy consumption.

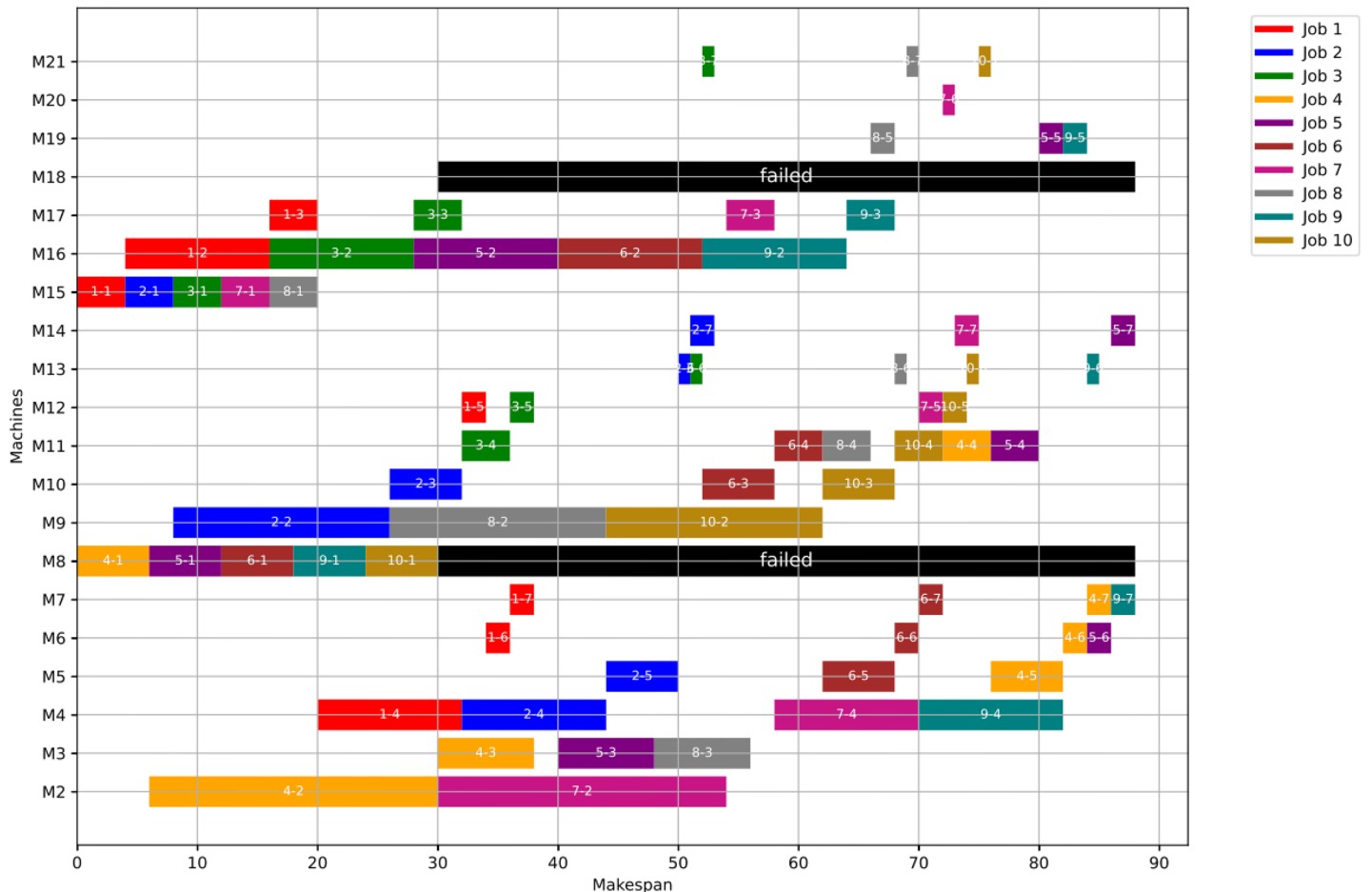

**Figure 26 Gantt chart with failure in M8 and M18 at disrupted time $T_F = 30$: case study (3) of Scenario 2.**

**Table 7 Starting time and completion time for all jobs before and after the failure in M1 and M10 (case study 1).**

| Healthy state | | | Scenario 1 | | | | | Scenario 2 | | |
|---|---|---|---|---|---|---|---|---|---|---|
| Job | Start time | Completion time | Job | Start time GWO | Completion time GWO | Start time PSO | Completion time PSO | Job | Start time | Completion time |
| 1 | 0 | 39 | 1 | 0 | 31 | 0 | 36 | 1 | 0 | 87 |
| 2 | 0 | 41 | 2 | 0 | 41 | 0 | 45 | 2 | 0 | 85 |
| 3 | 4 | 53 | 3 | 4 | 49 | 4 | 57 | 3 | 4 | 53 |
| 4 | 8 | 66 | 4 | 8 | 57 | 8 | 58 | 4 | 8 | 66 |
| 5 | 12 | 59 | 5 | 12 | 65 | 12 | 61 | 5 | 12 | 59 |
| 6 | 6 | 64 | 6 | 6 | 67 | 6 | 64 | 6 | 6 | 54 |
| 7 | 12 | 73 | 7 | 12 | 75 | 12 | 72 | 7 | 12 | 73 |
| 8 | 18 | 69 | 8 | 18 | 76 | 18 | 81 | 8 | 18 | 69 |
| 9 | 24 | 75 | 9 | 24 | 76 | 24 | 83 | 9 | 24 | 75 |
| 10 | 30 | 76 | 10 | 20 | 80 | 20 | 83 | 10 | 30 | 76 |

**Table 8 Comparison of the optimized makespan.**

| Case study | Makespan time | | | |
|---|---|---|---|---|
| | Healthy | Scenario 1 with GWO | Scenario 1 with PSO | Scenario 2 |
| 1 | 76 | 80 | 83 | 87 |
| 2 | 76 | 85 | 86 | 81 |
| 3 | 76 | 82 | 83 | 88 |

(a)        (b)        (c)

**Figure 27 Makespan extension in the two scenarios comparing the fault in the alternatives, (A) the miller machine, (B) the adding liquid machine, and (C) the cooler machine.**

**Table 9 Comparison of energy consumption before and after failure.**

| Example no. | Operation mode | Makespan | Energy consumption |
|---|---|---|---|
| Case study 1 | Healthy state | 76 | 529.61 *KwH* |
| | Scenario 1 GWO | 80 | 691.04 *KwH* |
| | Scenario 1 PSO | 83 | 563.23 *KwH* |
| | Scenario 2 | 87 | 529.61 *KwH* |
| Case study 2 | Healthy state | 76 | 580.59 *KwH* |
| | Scenario 1 GWO | 85 | 838.84 *KwH* |
| | Scenario 1 PSO | 86 | 625.03 *KwH* |
| | Scenario 2 | 81 | 590.07*KwH* |
| Case study 3 | Healthy state | 76 | 560.67 *KwH* |
| | Scenario 1 GWO | 82 | 814.35 *KwH* |
| | Scenario 1 PSO | 83 | 523.93 *KwH* |
| | Scenario 2 | 88 | 557.93 *KwH* |

Table 9 is provided to the decision makers in the manufacturing plant through a user dashboard, enabling them to evaluate the trade-off between time and energy consumption. Then, they select a balanced scheduling strategy based on additional factors such as product delivery deadlines, worker shift schedules, and the available budget.

The provided results illustrate the performance of different rescheduling strategies under three case studies, focusing on makespan and energy consumption. In the healthy

state across all three cases, the system consistently achieves the shortest makespan of 76-time units. This is expected, as a healthy state reflects normal operations without any disruptions. Additionally, energy consumption in this state is relatively low, establishing a baseline for evaluating the other scenarios.

When a machine breakdown occurs, Scenario 1 employs optimization techniques (GWO and PSO) to reschedule the operations. These methods inevitably lead to an increase in makespan due to the constraints imposed by the disrupted machine. In most instances, GWO produces a slightly shorter makespan than PSO, suggesting that it is more effective in minimizing total schedule time. However, this benefit comes with a cost: GWO tends to consume significantly more energy compared to PSO. For instance, in Case Study 1, GWO results in a makespan of 80 but with an energy consumption of 691.04 kWh, whereas PSO yields a makespan of 83 with only 563.23 kWh. This trend is especially prominent in Case Study 3, where GWO consumes 814.35 kWh, while PSO consumes just 523.93 kWh, highlighting PSO's superior energy efficiency despite slightly longer processing times.

Scenario 2 represents a simpler or more conservative rescheduling strategy. Although it consistently results in the highest makespans across all case studies, its energy consumption remains very close to that of the healthy state. This indicates that Scenario 2 likely involves minimal changes to the original schedule, thereby preserving energy efficiency but at the expense of longer completion times.

In summary, GWO is more aggressive in minimizing makespan but consumes more energy, while PSO strikes a better balance by controlling energy use at the cost of a marginally higher makespan. Scenario 2, on the other hand, maintains energy efficiency by limiting the extent of rescheduling, though this results in suboptimal makespan performance. The choice of rescheduling strategy, therefore, depends on whether the priority is to save time, reduce energy consumption, or maintain a balance between the two.

## CONCLUSIONS

In this study, we proposed a dynamic flexible job shop rescheduling framework that integrates two metaheuristic optimization algorithms—GWO and PSO—to efficiently reschedule operations following unexpected breakdown events. The approach was implemented within a digitally modelled industrial environment, aiming primarily to minimize the makespan of the production process.

The performance of the GWO and PSO algorithms was validated using standard benchmark datasets of varying sizes, demonstrating their effectiveness when compared to existing solutions in the literature. Three operational states were considered: the healthy state, Scenario 1 (optimized rescheduling using GWO or PSO), and Scenario 2 (rule-based rescheduling). Quantitative results confirm that both GWO and PSO significantly reduce the makespan in the healthy state. Furthermore, Scenario 1 consistently outperforms Scenario 2 in terms of makespan reduction, except in specific instances where failures occur on machines with short processing times, where Scenario 2 may yield slightly better results.

As anticipated, energy consumption was lowest in the healthy state. Between the two fault-handling strategies, Scenario 2 generally resulted in lower energy consumption but at the cost of a longer makespan.

A comparative assessment indicates that GWO achieves stronger makespan reduction but requires more energy, whereas PSO provides a compromise by lowering energy consumption with only a marginally higher makespan.

A key limitation of the proposed framework is the increased computational burden associated with scaling to larger problem instances. Additionally, the current energy consumption estimation could be enhanced by incorporating a more comprehensive model that includes all machines in the production environment.

### Funding
This work was funded by Princess Nourah bint Abdulrahman University Researchers Supporting Project number (PNURSP2025R239), Princess Nourah bint Abdulrahman University, Riyadh, Saudi Arabia. The funders had no role in study design, data collection and analysis, decision to publish, or preparation of the manuscript.

### Grant Disclosures
The following grant information was disclosed by the authors:
Princess Nourah bint Abdulrahman University, Riyadh, Saudi Arabia: PNURSP2025R239.

### Competing Interests
The authors declare that they have no competing interests.

### Author Contributions
- Nehal Tarek conceived and designed the experiments, performed the experiments, analyzed the data, performed the computation work, prepared figures and/or tables, authored or reviewed drafts of the article, and approved the final draft.
- Samia Allaoua Chelloug conceived and designed the experiments, authored or reviewed drafts of the article, and approved the final draft.
- Soha Alhelaly performed the experiments, authored or reviewed drafts of the article, and approved the final draft.
- Nancy A. El-Hefnawy conceived and designed the experiments, authored or reviewed drafts of the article, and approved the final draft.
- Hatem Abdel-Kader analyzed the data, authored or reviewed drafts of the article, and approved the final draft.
- Amira Abdelatey conceived and designed the experiments, analyzed the data, prepared figures and/or tables, authored or reviewed drafts of the article, and approved the final draft.

### Data Availability
The code is available in the Supplemental File.

## Supplemental Information

Supplemental information for this article can be found online at http://dx.doi.org/10.7717/peerj-cs.3379#supplemental-information.

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
