# Peer review of "Swarm algorithms for sustainable dynamic flexible job shop rescheduling under machine breakdown in smart manufacturing plants"

_PeerJ Computer Science, doi:10.7717/peerj-cs.3379_

## Round 0.1 · original submission · Major Revisions

· Academic Editor

Major Revisions

The paper requires substantial revision.

**Language Note:** The review process has identified that the English language must be improved. PeerJ can provide language editing services - please contact us at [email protected] for pricing (be sure to provide your manuscript number and title). Alternatively, you should make your own arrangements to improve the language quality and provide details in your response letter. – PeerJ Staff

Reviewer 1 ·

Basic reporting

This study proposes two machine failure handling strategies using the grey wolf optimization (GWO) algorithm. Both scenarios incorporate operational constraints and evaluate energy consumption to ensure efficiency. Both scenarios demonstrate effective rescheduling, with energy consumption analysis confirming rational energy use. However, the authors did not explain the specific application of DT technology in rescheduling, and only used the concept of DT. This study is actually to design a rescheduling algorithm based on GWO, which has no practical relationship with DT. In addition, there are many defects in the content of this article, and I suggest rejecting it.

1. For a research paper, keywords are essential, but I did not find them.

2. In the introduction, the author discusses JSP, FJSP, and the corresponding dynamic problems but fails to summarize the shortcomings of existing research and explain the motivation for this paper. Moreover, the current version of the introduction lacks logical coherence, making it unclear why the authors chose to study this issue.

3. The structure of this manuscript needs to be adjusted, and the author should briefly describe the work done and summarize the innovations in the introduction. By this, I mean that the list of main contributions in Section 2 should be placed at the end of Section 1.

4. The phrasing in sentences such as lines 77, 121, and 128 makes comprehension difficult.

5. The manuscript needs a more thorough literature review, particularly regarding previous work on machine failure handling in job shop scheduling problems and the application of DT technology in industrial rescheduling scenarios. However, the author does not introduce the related research on DT.

6. The normativity of this article needs to be further improved. The authors alternated between the full name and the short name, such as "digital twin".

7. Figure 3 is beautiful, but the colors are too rich, and I suggest redrawing it.

8. The authors directly provide the parameter values of the algorithm. Were all these parameters determined using Taguchi experiments?

9. The manuscript mentions energy savings but does not clearly show how these savings are calculated or how different scenarios contribute to energy optimization.

10. It would be helpful to compare the proposed GWO-based strategies with other well-established rescheduling techniques to strengthen the novelty of the approach.

11. At the same time, for the experimental results, the nonparametric statistical test method should be used to verify the effectiveness of the algorithm.

12. How is the virtual-real mapping technology of DT implemented in the proposed method? Furthermore, how does the author solve the delay problem between the digital model and the physical model? None of these are described in the paper.

Experimental design

-

Validity of the findings

-

Reviewer 2 ·

Basic reporting

1. This paper needs further polishing. Some sentences are very difficult to understand.
2. Regarding literature works, the authors reviewed some previous works on GWO for the production scheduling problem. However, authors have missed some related works. Various studies have a close relation with this topic, such as a hybrid multi-objective grey wolf optimizer for dynamic scheduling in a real-world welding industry, and a multi-objective cellular grey wolf optimizer for the hybrid flowshop scheduling problem considering noise pollution. The review of this paper will be further enriched if they are included.
3. Why adopt GWO rather than GA, PSO, and other metaheuristics to solve the flexible job shop scheduling problem with dynamic events?
4. In general, dynamic FJSP is a multi-objective optimization problem. Authors should consider production system stability or robustness in addition to the makespan objective.
5. The title of the paper is not clear. The algorithm is not clear, and the problem under study is not clear. It is DFJSP rather than DJSP.
6. How is sustainability reflected?

Experimental design

1. The performance of GWO is sensitive to the parameter setting of the metaheuristics. Thus, parameter setting should be calibrated.
2. The current experiments are insufficient to verify the effectiveness of your proposal.

Validity of the findings

In the experiment, statistical tests like the t-test should be conducted to validate the significant difference between different metaheuristics due to the statistical probability characteristics of these metaheuristics. Moreover, the deep reason why the proposal is better than its counterparts should be stated more clearly in the results analysis.

---

## Round 0.2 · Major Revisions

· Academic Editor

Major Revisions

Please incorporate the suggestions of the reviewer.

Reviewer 1 ·

Basic reporting

In this paper, two well-known meta-heuristic algorithms are used to solve the DFJSSP, much of the contribution lies in engineering implementation rather than scientific advancement. Although the authors have answered some of the questions, most of them have not been explained clearly, such as suggestions 2, 3, 5, 9, 11 and 12. My main opinions are as follows.
1. The title of the paper is too general. Neither evolutionary algorithms nor digital twins seem to have much relevance to the main text.
2. The discussion literature on the relevant work is not recently published, and it is recommended to use a table for summary.
3. Apart from the relevant work analysis, the methods and simulation experiments proposed by the authors seem to have no relation to digital twins.
4. There are still many variables in models and algorithms that lack definitions.

Experimental design

5. The performance of GWO and PSO under different problem sizes, parameter variations, or uncertainty levels is not analyzed.
6. While energy metrics are reported, the impact of algorithmic decisions on real power consumption is not validated or rigorously modeled.
7. The case study is restricted to one synthetic poultry factory dataset, which limits the generalizability of results.

Validity of the findings

no comment

Reviewer 2 ·

Basic reporting

All my concerned questions have been done.

Experimental design

All my concerned questions have been done.

Validity of the findings

All my concerned questions have been done.

Additional comments

It can be accepted.

---

## Round 0.3 · Minor Revisions

· Academic Editor

Minor Revisions

Incorporate the reviewer's comments.

Reviewer 3 ·

Basic reporting

no comment

Experimental design

no comment

Validity of the findings

no comment

Additional comments

no comment

Reviewer 4 ·

Basic reporting

In the revised version of the manuscript, the topic is communicated well, mostly using proper English. Small language errors remain though and should be fixed for a final version. The introduction explains the topic properly and cites relevant literature for background information, the used algorithms, etc. For completeness, I would expect a footnote with a link to the mentioned "AI and Localizatiopn of Industry in Egypt" though.

Mistakes (example):
"A GWO and a PSO algorithms" should be either "A GWO and a PSO algorithm" or "GWO and PSO algorithms"
"have been developed" -> "have been applied" (GWO and PSO already existed as algorithms beforehand)

Related Work:
Small inconsistency: some authors are mentioned by name ("Wang et al.", "Baykasoğlu") while others are just referred to as "the authors in [30]".

Experimental design

The experiments build upon the Brandimarte benchmark set with three case studies used to evaluate the two algorithms. This is well-defined and reproducible.

Validity of the findings

The findings appear to be valid, but should be communicated better. Neither the abstract nor the conclusion give a hint about the performance differences of the two algorithms, although using those in the context of a job shop scheduling setup is the main theme of the paper. The last paragraph in Section 5 gives an assessment of the different behavior of the algorithms, please consider reporting this information in brief also more prominently in abstract and conclusion.

---

## Round 0.4 · accepted · Accept

· Academic Editor

Accept

The paper may be accepted.

Reviewer 4 ·

Basic reporting

The updates in the paper address the previous concerns about language mistakes well.

Experimental design

As already indicated in the last review, the experiments build upon the Brandimarte benchmark set with three case studies used to evaluate the two algorithms. This is well-defined and reproducible.

Validity of the findings

The updates in the paper for communicating the outcome in the abstract and conclusion have addressed the previous concerns well.